# Intermediate-state-trapped mutants pinpoint G protein-coupled receptor conformational allostery

Xudong Wang [1,5], Chris Neale [2,5], Soo-Kyung Kim [3], William A. Goddard [3] & Libin Ye [1,4] ✉

Understanding the roles of intermediate states in signaling is pivotal to unraveling the activation processes of G protein-coupled receptors (GPCRs). However, the field is still struggling to define these conformational states with sufficient resolution to study their individual functions. Here, we demonstrate the feasibility of enriching the populations of discrete states via conformation-biased mutants. These mutants adopt distinct distributions among five states that lie along the activation pathway of adenosine $A_{2A}$ receptor ($A_{2A}R$), a class A GPCR. Our study reveals a structurally conserved cation-π lock between transmembrane helix VI (TM6) and Helix8 that regulates cytoplasmic cavity opening as a "gatekeeper" for G protein penetration. A GPCR activation process based on the well-discerned conformational states is thus proposed, allosterically micro-modulated by the cation-π lock and a previously well-defined ionic interaction between TM3 and TM6. Intermediate-state-trapped mutants will also provide useful information in relation to receptor-G protein signal transduction.

The structural characterization of GPCRs by X-ray crystallography and cryogenic electron microscopy (cryo-EM) has provided unprecedented insights into the processes of receptor activation[1–6]. However, our comprehension of GPCR signaling remains limited because these techniques typically capture low-energy states, whereas the full articulation of diversity and complexity in GPCR signaling requires descriptions based on dynamic ensembles in which conformational changes are continuous. Based on existing structures[3,4,6,7], a typical GPCR signaling pathway can be defined to start with agonist binding, which shifts the receptor's conformational equilibrium to favor active forms in which the cytoplasmic cavity opens and becomes more permissive to a deep Gαβγ binding. In response to the receptor's conformational change, the C-terminal α helix of the Gα protein (Cα5) undergoes a conformational rearrangement and penetrates the receptor's cytoplasmic cavity. Subsequently, Cα5 binds the receptor more tightly while the Ras-like (RS) domain and the α-helical domain (AHD) of the G protein separate from one another. This separation facilitates GDP release followed by GTP binding to the widened Gα cavity (i.e., nucleotide exchange), which triggers a series of downstream signaling events[8] (Supplementary Fig. 1). However, the mechanisms by which the conformational state of the receptor is sequentially activated and then adapted for intracellular Gαβγ binding remain elusive.

Previous research has reported a regulatory "ionic lock" ($DR^{3.50}Y$-$D/E^{6.30}$) (superscript denotes Ballesteros-Weinstein numbering[9]) form a tight interaction between transmembrane helix III (TM3) and TM6[10,11]. Despite the suggestion that this "ionic lock" plays a crucial role in stabilizing inactive receptor states, it is broken (i.e., not formed) in several inactive-state crystal structures, including those of $β_1$ adrenergic receptor ($β_{1A}R$), $β_2AR$, and $A_{2A}R$ receptors[12–16].

[1]Department of Cell Biology, Microbiology and Molecular Biology, University of South Florida, Tampa, FL 33620, USA. [2]Theoretical Biology and Biophysics, Los Alamos National Laboratory, Los Alamos, NM 87545, USA. [3]Materials and Process Simulation Center (139-74), California Institute of Technology, Pasadena, CA 91125, USA. [4]H. Lee Moffitt Cancer Center & Research Institute, 12902 USF Magnolia Drive, Tampa, FL 33612, USA. [5]These authors contributed equally: Xudong Wang, Chris Neale. ✉e-mail: libinye@usf.edu

(Supplementary Fig. 2). This finding suggests that the ionic lock between TM3 and TM6 may more directly regulate microswitches within and between distinct inactive states rather than regulating transitions between inactive and active conformations. This interpretation leads us to question which micro-electrostatic switches regulate the crucial conformational transitions between inactive and active states, and thus the whole sequential activation process is regulated by these micro-switches.

In this work, with the guidance of computational simulations, we create a set of conformation-biased constructs using site-directed mutagenesis. Their conformational profiles are probed by $^{19}$F NMR, along with conformational transitions among different states that lies in the activation process. Through these investigations, we identify a conserved "cation-π" toggle switch between R291$^{7.56}$/R293$^{8.48}$ on TM7/H8 and H230$^{6.32}$ on TM6, serving as a "gatekeeper" to modulate the G protein insertion to the intracellular G protein binding cavity. This switch also regulates the conformational transitions between the inactive conformational ensembles and the active conformatioal ensembles, different from the previously identified ionic lock between the TM3 and TM6 that regulates the conformational transitions between two inactive states.

## Results

With the help of molecular dynamics (MD) simulation of the detergent solubilized receptor and molecular docking, we generated models of A$_{2A}$R-Gα$_s$βγ bound to either a partial agonist (LUF5834) or a full agonist (NECA)$^{17}$, representing partially activated A$_{2A}$R-Gα$_s$βγ (Fig. 1b) and fully activated A$_{2A}$R-Gα$_s$βγ complexes (Fig. 1c), respectively, in addition to a pre-coupled complex (Fig. 1a). The NECA-A$_{2A}$R-Gα$_s$βγ model reveals two intermolecular salt bridges between intracellular loop 3 (ICL3) in the A$_{2A}$R and the AHD of Gα$_s$ (Fig. 1c). Importantly, these salt bridges are lost in the predicted NECA-A$_{2A}$R-Gα$_s$βγ complex (Fig. 1b). This ligand-specific behavior inspired us to design receptor mutations that could quench the signal from the full agonist, with expectation of eliminating the corresponding conformational state and thus stabilizing intermediate states. To our surprise, the mutants based on these presumptions created a series of conformation-biased constructs, including those trapped intermediate states. These advances provided

insights into the roles of R291 and R293 in the receptor activation process through allosterically modulating the opening of G protein binding cavity.

Previously, we established a $^{19}$F-NMR system for probing the A$_{2A}$R, in which two active and two active-like conformational states were delineated in the detergent micelles and lipid nanodiscs using T2-based spectral deconvolution$^{16,18–20}$. However, T2 measurements are limited by difficulties in distinguishing overlapping states since they represent a convolution of three independent values (chemical shift, linewidth, and intensity). Here, we mitigate this limitation with point mutations in TM7/H8 that change the profile of stability among detergent solubilized receptor states. We also reduced the detergent MNG-3 concentration to 0.02% compared to previous studies (0.1%). A lower concentration of detergent has been reported to provide a better NMR resonance resolution for dodecylmaltoside (DDM) solubilized rhodopsin$^{21}$, and our present study indicates that maltose-neopentyl glycol/cholesteryl hemisuccinate (MNG-3) has a similar property to the DDM. These advances facilitate more precise characterizations of intermediate states and their roles in activation pathways. Thus, the receptors in this manuscript will be reconstituted in the MNG-3/CHS system$^{19,22}$.

Specifically, we reexamined the conformational progression from the inactive state (S1) to the fully activated state (S5) for the A$_{2A}$R receptor in MNG-3 detergent with CHS, as shown in Fig. 2a, using a probe at V229C$^{6.31}$ on the TM6 helix$^{23,24}$. Because TM6 rotates out and exposes residue 229 to the aqueous environment during the receptor activation process, these NMR spectra delineate conformational states based on differential solvent exposure, in which the five-state S1-to-S5 progression is ordered from the low magnetic field (less aqueous exposure) to the high magnetic field (more exposure) (Fig. 2b, c). Here, S5, which represents the fully activated conformational state, was defined by the addition of full agonist NECA, Gα$_s$βγ, Mg$^{2+}$, and GDP (Fig. 2a, the bottom spectrum). This is consistent with the previous studies showing that protein $^{19}$F NMR signals tend to shift downfield and broaden as the $^{19}$F probe encounters a more hydrophobic (less aqueous) environment$^{25,26}$. The populations of each conformational state can also be calculated based on the integrals of each delineated resonance, as shown in Fig. 2d.

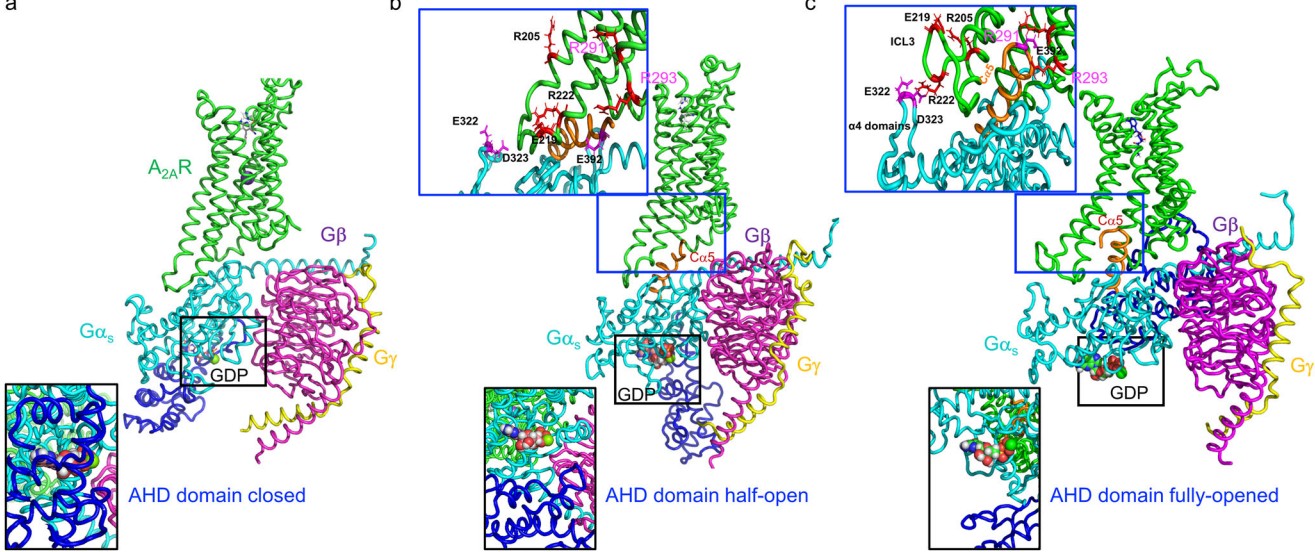

**Fig. 1 | The simulated A$_{2A}$R complex with partial and full agonists that exhibited different conformational states of G proteins, in which the AHD domain was either partially or fully opened. a** Pre-coupled complex of A$_{2A}$R-Gα$_s$βγ; A$_{2A}$R in green, Gα$_s$ in cyan (AHD domain in blue), Gβ in magenta, and Gγ in yellow. **b** Partial agonist LUF5834-bound A$_{2A}$R-Gα$_s$βγ complex showing interfacial interaction between the A$_{2A}$R and Gα$_s$ with a partially open AHD domain. **c** Full agonist NECA-bound A$_{2A}$R-Gαβγ complex showing interfacial interaction between the A$_{2A}$R and Gα$_s$ with a fully opened AHD domain. Residues of R291$^{7.56}$ and R293$^{8.48}$ are highlighted.

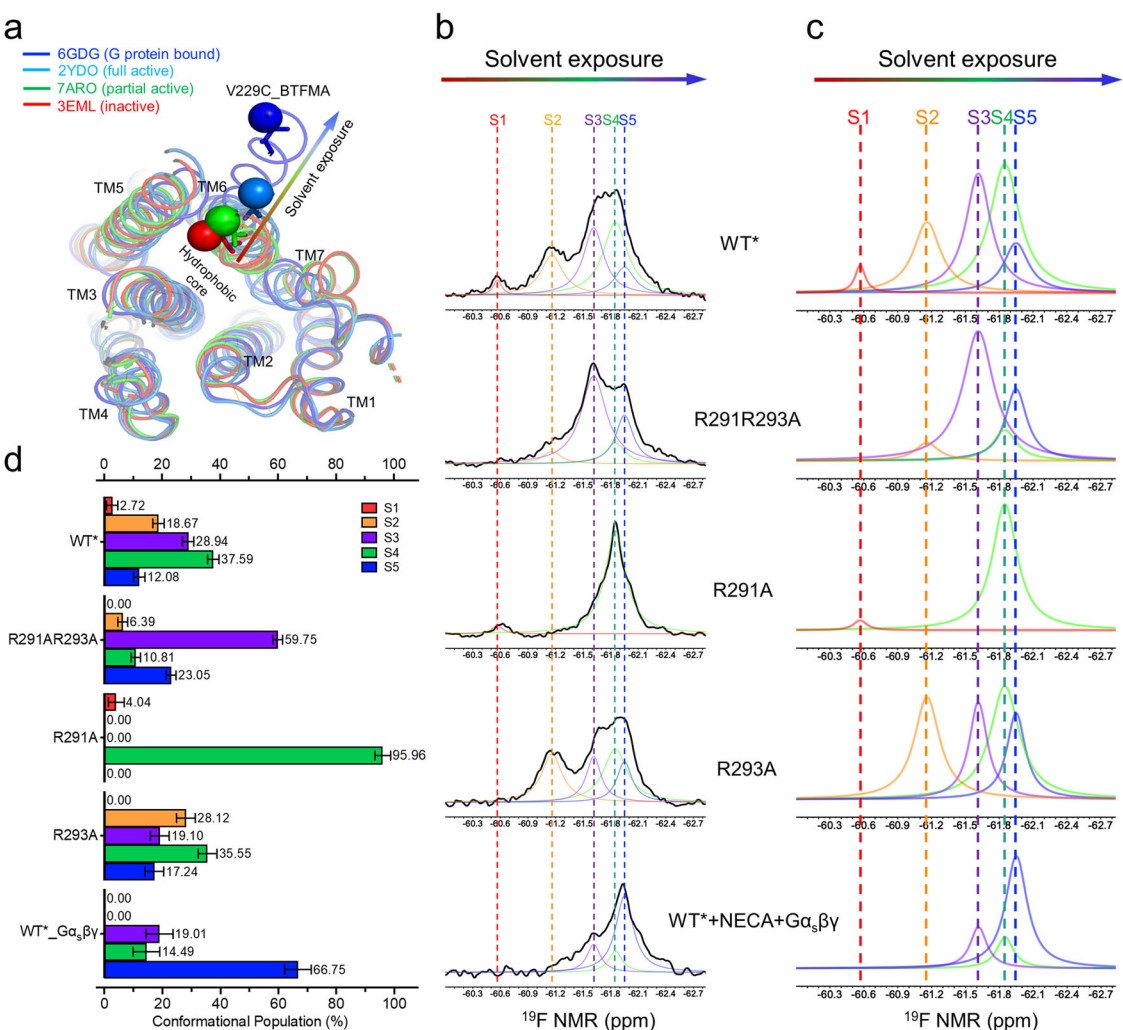

**Fig. 2 | Conformation-biased mutants probed by NMR through a ¹⁹F label on the residue V229C. a** Overlay of crystal structures indicating the position of the BTFMA labeled residue 229 on TM6 reveals increased solvent exposure during activation. **b** Conformational profiles probed by ¹⁹F-BTFMA labeled NMR experiments for each mutant and WT* in apo state, in which the spectrum of NECA + WT* + Gαβγ serving as the reference as the fully activated state S5. **c** The deconvoluted ¹⁹F NMR spectra for from (**b**); S1 state in red, S2 state in orange, S3 state in purple, S4 state in lime, and S5 in blue. **d** Population distributions of each conformational states in different conformation-biased mutants and WT*; the source data for these population distributions are included in the Source Data file. Data with error bars are presented as state population±SD. The SD values were determined based on spectral S/Ns and fitting errors of the deconvolutions.

As indicated in Fig. 2b, c, the R291A mutant allowed us to "physically" define the S4 state that wasn't able to be unambiguously distinguished from others without NMR spectral deconvolution in the previous research[18,19,27]. The R291AR293A double mutant allowed us to better reveal discrete states S3 and S5 as well, along with a small portion of the S4 state. The chemical shifts of each conformational state could thus be clearly defined for the apo sample. In contrast, the chemical shifts of each state in the previous studies were largely based on the spectral deconvolution or fitting process. Of note, the T2-based measurement was also used to validate the linewidths of each conformational component, as shown in Supplementary Fig. 3, to support the deconvolutions we performed here based on conformation-biased constructs. Considering three variances for each conformational state in the deconvolution process that lead to the complexity of spectral fitting, we defined the chemical shifts of each state using the values presented in these conformation-biased mutants with ±0.05 ppm variations for each state during the fitting process, considering conformational dynamics in varied samples. The linewidth and intensity of each state are then determined through the best fitting of each ¹⁹F NMR spectrum with a minimal fitting error, which serves as a part of standard deviation for each conformation component shown in

Fig. 2d. The addition of NECA or ZM241385 removed the S1 state, but the partial agonist LUF5834 did not otherwise dramatically perturb the apparent conformational equilibrium of R291A (Fig. 3a).

In contrast to the R291A mutant, the apo state of the R291AR293A double mutant predominantly adopted S3 and S5 states with an invisible portion of the S4 that is only discernible with the assistance of spectral deconvolution (Fig. 2 and Fig. 3c). Furthermore, the S4 state was not preferentially stabilized by any of the tested ligands, including the partial agonist LUF5834 (Fig. 3c). The ¹⁹F NMR spectra clearly indicate that the double mutant R291AR293A stabilizes S3 and S5 at the expense of other states, further supporting the pivotal role of R293[8,48] in maintaining the S4 and S1 states. Conversely, R291[7.56] may be involved in maintenance of the S2 state. The addition of the full agonist NECA or the inverse agonist ZM241385 to this double mutant shifts the spectrum to the S5 or S3 states, respectively. However, this double mutant was unable to rebalance its equilibrium to S1 and S2 states even though a small population of S2 was observed in inverse agonist ZM241385-saturated samples (bottom spectrum, Fig. 3c). Combining these results, we hypothesize that bundle packing between TM6 and TM7/H8 is critical in maintaining the inactive conformational states S1 and S2, though previous studies had indicated the TM6 and TM7/H8

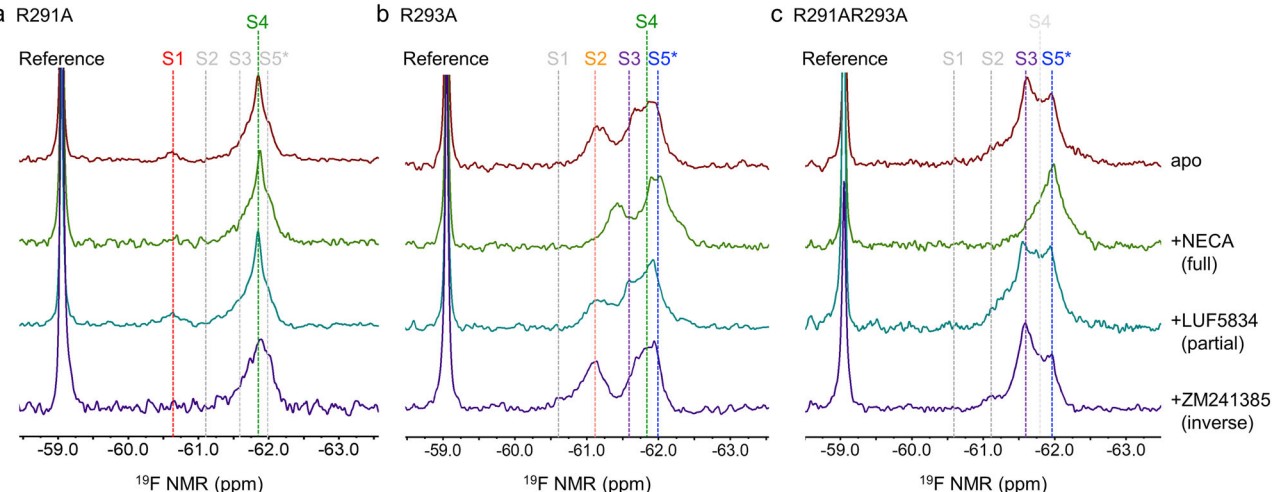

**Fig. 3 | ¹⁹F NMR spectra of mutants as a function of ligands.** Data show ¹⁹F-BTFMA labeled **a** R291A, **b** R293A, and **c** R291AR293A without ligand and in presence of full agonist NECA, partial agonist LUF5834, and inverse agonist ZM241385. Of note, apo in maroon, apo+NECA in green, apo+LUF5834 in teal, apo+ZM241385 in violet.

bundle was involved in stabilizing the inactive states[28–30], in which two inactive states concurred with differential H8/TM7 movements[31,32]. This could occur when both TM6 and TM7/H8 adopt inactive or partially activated poses, considering the rotations of TM6 and TM7/H8 during the activation process and the possibility of transient helical rejoining during the activation process. This was further validated by NMR spectra for the R291A mutant, where S1 and S4 states co-exist, and ionic lock DR$^{3.50}$Y-E$^{6.30}$ formation between TM3 and TM6, was expected for the partial agonist bound receptor. From this point, the receptor is either further activated from the S4 state to the S5 state upon the addition of full agonist or moved from the S4 state to the S3/S2 states upon the addition of inverse agonist. In both cases, the ionic lock appears to break and TM6 may begin to separate from TM7/H8, as suggested by the R291A spectra with either NECA or ZM241385, where the S1 state disappeared. Conversely, the S1 state was maintained with the addition of LUF5834, as shown in Fig. 3a. While an ensemble of twenty-seven 1 μs MD simulations was unable to quantify state transitions with statistical significance, sampled configurations were consistent with activation pathways that involve both ionic lock breakage and helix 6 rotation (Supplementary Fig. 4). Specifically, simulations starting from inverse agonist- and partial agonist-bound conformations remained in inactive states with ionic lock flickering and helix 6 rotated to expose H230 and partially bury V299, whereas simulations starting from an adenosine-bound conformation usually maintained a broken ionic lock with helix 6 rotated to more fully expose V229 to aqueous solution. These results also support the involvement of R291$^{7.56}$ and R293$^{8.48}$ in stabilizing states S2 and S1, respectively. Simulated micelle shape and detergent-protein contacts are shown via a representative structure and time- and ensemble-averages in Supplementary Fig. 5.

As aforementioned, the double mutant R291AR293A, predominantly adopts S3 and S5, with a small portion of the S4 state (Fig. 2 and Fig. 3). Importantly, NECA drove a greater S5 population in this mutant, suggesting that it remains capable of being "physically" activated with a fully opened the G protein binding cavity (Fig. 3c). Interestingly, NMR spectra for R291A revealed that a shift to the fully activated state may not be necessary for the receptor to trigger intracellular signaling, given that R291A did not accumulate the S5 state significantly even with the addition of NECA.

In comparison to WT* (defined as the construct of A$_{2A}$R_316_V229C and all other mutants in this manuscript were mutated based on this construct), the isolated R293A mutation leads to the disappearance of the S1 state in the apo sample, while the S4 state in this mutant is overlapped by S3 and S5 and therefore difficult to evaluate prior to the addition of the partial agonist, LUF5834, which stabilizes S4 (Fig. 3b). Conversely, inactive states S2 populated upon the addition of the inverse agonist ZM241385, while the full agonist NECA drove the conformational equilibrium to the fully activated state S5. All receptors were evaluated for functionality via radioligand binding assays (Supplementary Fig. 6a) and for purity via SDS-PAGE (Supplementary Fig. 6b).

Structure and sequence analyses suggest that a toggle switch between R291$^{7.56}$/R293$^{8.48}$ on TM7/H8 and H230$^{6.32}$ on TM6 may be involved in the differential stabilization of various receptor states outlined above and regulating the G protein. This would serve as a regulator of TM6 rotation and thereby cytoplasmic cavity opening for G protein binding, as shown in Fig. 4a. To this point, we hypothesize that R293$^{8.48}$ has a "gatekeeper"-like functional motion in which it swings back and forth to regulate the closing and opening of the cavity. Analyses of representative crystal structures support this hypothesis since R293$^{8.48}$ rotates by 360° during activation (Fig. 4b). We also assumed that the loss of R293 by mutating it to A residues would facilitate the opening of the G protein binding pocket and lead to a higher receptor activity than the WT*. As shown in Supplementary Fig. 7, the GTPase hydrolysis was used for preliminarily evaluating the regulation capacity of each mutant on the GTP-GDP conversion. The data indicated that the mutation of R291 to A residues resulted in a significant hydrolysis activity decrease in both mutants of R291A and R291AR293A whereas the single mutation of R293A slightly increased the activity of the receptor. Combining the NMR data stated above, this observation implies that the residue of R291 is involved in not only the opening of the G protein binding cavity but also the regulation of the GTP hydrolysis of G proteins. In contrast, R293 mainly serves as a gatekeeper to modulate the Cα5 insertion of G protein because the mutant R293A only exhibited a slightly higher activity than the WT*. Of note, the mutant of R293A maintains a similar conformational profile to the WT* except for the loss of a small portion of the S1 state.

We further hypothesize that the conformational changes described in Fig. 2 are mediated by cation-π interactions between R291$^{7.56}$/R293$^{8.48}$ and H230$^{6.32}$ from TM6, along with the previously reported ionic lock DR$^{3.50}$Y-E$^{6.30}$ between TM3 and TM6. Specifically, we propose the toggle forces from ionic lock DR$^{3.50}$Y-E$^{6.30}$ and cation-π locks R291$^{7.56}$/R293$^{8.48}$-H230$^{6.32}$ regulate the motion of TM6, resulting in various degrees of opening the intracellular pocket for G protein binding and thereby, differential signaling efficacies. Structure-based

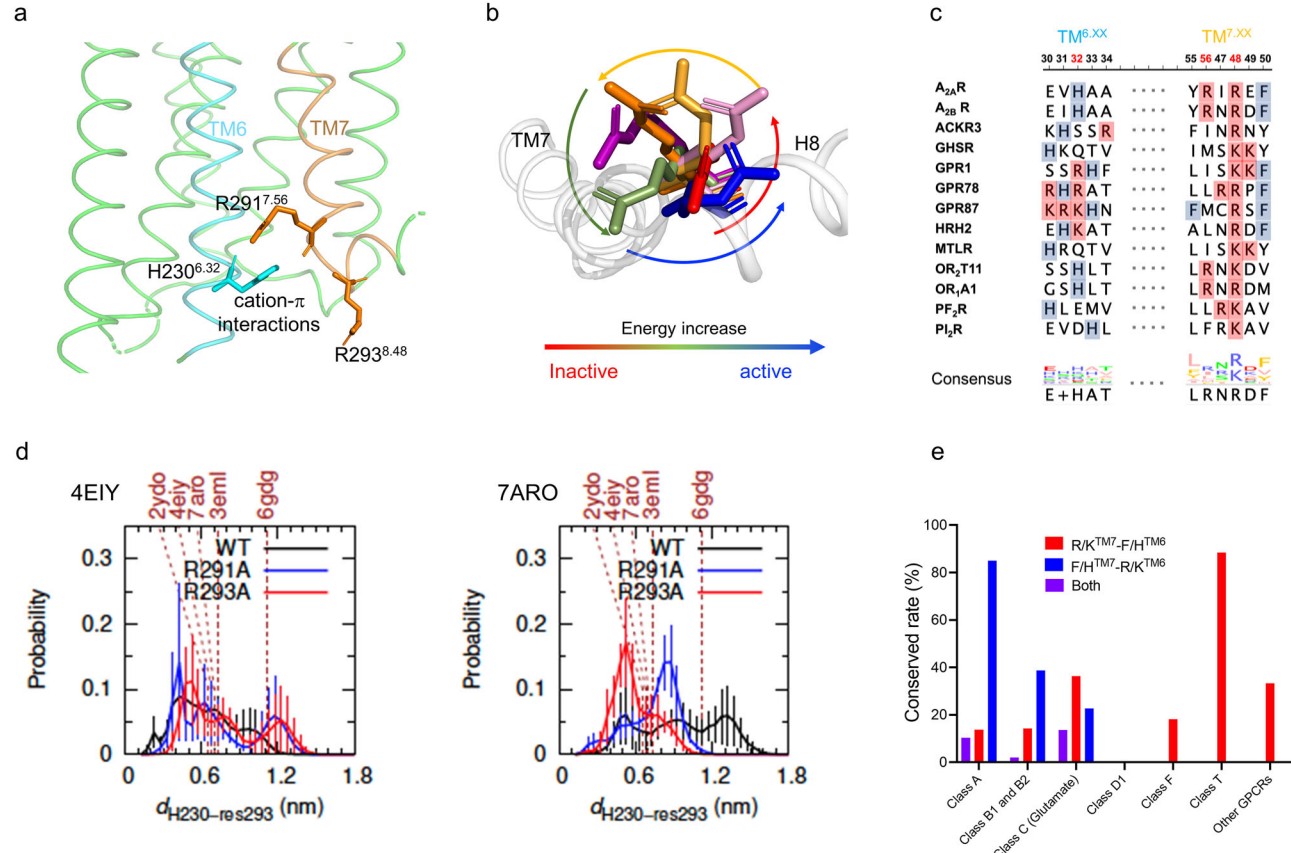

**Fig. 4 | Cation-π interactions between TM6 and TM7/H8. a** Topological view of cation-π interaction involving R291[7.56]/R293[8.48] and H230[6.32] in TM6 (cyan) and TM7/H8 (orange), respectively. **b** Cytosolic view of R293[8.48] rotation in representative structures. **c** Sequence-based alignment depicting R291[7.56]/R293[8.48]-H230[6.32] in different GPCRs; the source data for these alignments are included in the Source Data file. **d** H230[6.32]-R293[8.48] interaction probability in simulations of 7ARO, 4EIY, and their mutants. The dash lines stand for the distance in the resolved structures. 0.6 nm is an effective distance for cation-π interaction. **e** Potential conservation rates of cation-π interaction between TM6 and TM7/H8 cross families using structure-based alignments.

alignment using GPCRdb[33] indicates a potentially high conservation of the residues mediating the toggle force between TM6 and TM7/H8 across GPCR families except for family D1 (Fig. 4c, e, and Supplementary Fig. 8), though it is only ~8% conserved when a strict sequence-based alignment is conducted, even in the family A.

Provided that H230[6.32] is involved in such a cation-π interaction, we expect that an H230A mutation would not populate either S1 or S2 states, considering that interactions between R291[7.56]/R293[8.48]-H230[6.32] are involved in the stabilization of these two states. As shown in Supplementary Fig. 9, the H230A point mutant led to the expected NMR spectrum. Therefore, this orthogonal mutation is consistent with a role for R291[7.56]/R293[8.48]-H230[6.32] interactions in regulating conformational transitions between the inactive ensembles (S1 and S2) and active-like ensembles (S3, S4, and S5). Our MD simulations also indicate that direct interaction between R293[8.48] and H230[6.32] occurs more frequently when starting from the WT-7ARO than from WT-2YDO (Supplementary Fig. 10a, c) with probabilities ranging from 0.22 to 0.60. This simulation data is consistent with the existence of an R293[8.48]-H230[6.32] interaction that breaks during activation. Conversely, simulations indicate that the R291[7.56]-H230[6.32] interaction is more stable and less sensitive to receptor activation (Supplementary Fig. 11). Of note, effective cation-π interactions are usually within 0.6 nm between component residues[34] (Supplementary Fig. 10b).

## Discussion
Combined with previous reports[18,24] and the ¹⁹F NMR spectra discussed here, dynamic transition processes among all different conformational states are depicted in Fig. 5. In the S1 state, both ionic lock DR[3.50]Y-E[6.30] and cation-π interactions (R291[7.56]-H230[6.32], and R293[8.48]-H230[6.32]) are formed, with TM3, TM6, and TM7/H8 tightly packed together. As activation proceeds, the S1 state transitions to the S2 state, where the ionic lock (DR[3.50]Y-E[6.30]) is broken. At this stage, as shown for the apo-R293A mutant (Fig. 2), the S2 state remains highly populated, indicating that R293[8.48] is not required for S2 maintenance when R291[7.56]-H230[6.32] is still locked in position but highly related to the S1 state formation. The loss of interactions between R291[7.56] and H230[6.32] appears to destabilize the S2 state, as reflected in both R291A and R291AR293A mutants. R293[8.48] is required for maintaining the S4 state, suggesting that its further disengagement with H230[6.32] will lead the receptor toward the complete open of G protein binding cavity, as evident in the R291AR293A mutant spectrum. Because ¹⁹F NMR profiles reveal that state populations and exchange dynamics depend on environment (e.g., detergent micelle vs. lipid nanodisc)[18,19,35], additional studies are necessary to assess the relative importance of the S4 state of the A₂ₐR in lipidic environments.

Due to the challenges of GPCR conformational delineation and intrinsic receptor plasticity, a complete sequential GPCR activation process is difficult to establish. To achieve this goal, a resolvable conformational profile is required. ¹⁹F represents one of the most sensitive nuclei to microenvironmental changes and it has been utilized to profile the conformational states of GPCRs for many years[36]. However, due to the large chemical shift anisotropy effect (CSA) of macromolecular ¹⁹F NMR, conformational states often suffer from spectral overlap resulting from linewidth broadening. This has

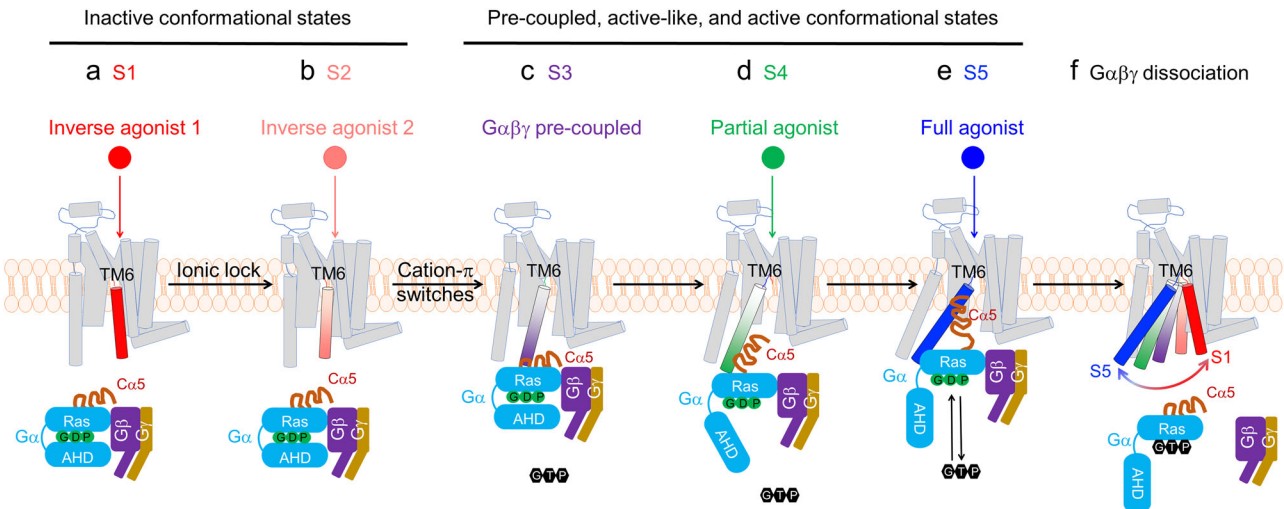

**Fig. 5 | The allosteric roles of ionic lock between TM3 and TM6 and cation-π microswitches between TM7/H8 and TM6 in the A2AR activation. a, b** In the inactive states the GDP-loaded Gαβγ and GPCR do not interact. The receptor undergoes a transition between two inactive states featured with ionic lock (DRY-E) switch between the helices TM3 and TM6. **c** Pre-coupled complex. The receptor and Gαβγ are evidently pre-coupled prior to be activated. **d** Upon partial agonist bindings, the receptor undergoes conformational rearrangement to form the S4 state to facilitate Cα5 to penetrate the binding cavity. **e** Upon the full agonist binding, the GPCR activation continues to proceed until the fully opened conformational complex is formed between the receptor and Gαβγ, resulting in a free nucleotide exchange. **f** After GTP binding, the Gαβγ will disengage the receptor and dissociate into the Gα and Gβγ. Of note, the allosteric switch from the inactive states to pre-coupled, active-like, and active states were modulated by the cation-π interactions between the TM7/H8 and TM6, in which R291 and R293 residues play critical roles in controlling the G protein cavity opening and regulating G protein hydrolysis.

historically occurred with both BTFMA, and TET labeled A$_{2A}$R[18,19,37]. Furthermore, differences in chemical probes, sample preparation procedures, and solubilizing amphiphiles can affect the resulting $^{19}$F spectra of A$_{2A}$R[35] and β$_2$AR[19,24,38]. The current research reduces the ambiguity in the conformational profiles caused by spectral overlap for the A$_{2A}$R via conformation-biased mutants, while also providing more molecular details of how the micro-switches along and among transmembrane helices allosterically drive the receptor activation (Fig. 5). These mutants will facilitate further studies of intermediate complexes in GPCR signaling, including resolving the intermediate complex structures using intermediate state trapped mutants. This would provide the additional information necessary to define receptor activation beyond the simple two-state model. It is also worthy of note that because $^{19}$F profiles reveal that state populations and exchange dynamics depend on the environment (e.g., detergent micelle vs. lipid nanodisc), additional studies are necessary to assess the relative importance of the intermediate states of the A$_{2A}$R, such as the S4 in lipidic environments and if it can be successfully populated in it as well, considering that the HDL systems prefer to shift the conformational equilibrium to the fully activated states as aforementioned[19,27]. This advance also has the potential to guide drug design based on distinct GPCR and G protein responses to various ligands[39,40], for instance by targeting drug leads that favor distributions of S3, S4, and S5. We expect that this advance will also be more broadly applicable for trapping or stabilizing functional or transition states in other proteins, such as catalytic enzymes, with conformation-biasing point mutations[41].

## Methods
### Plasmid construction and transformation
The full-length human A$_{2A}$R gene, originating from construct pPIC9K_ADORA2A[1], was generously provided by Prof. Takuya Kobayashi (Kyoto University, Kyoto, Japan). The construct A$_{2A}$R_316, constructed in our previous study, has an integrated FLAG tag on the N-terminus and a poly-his tag on the C-terminus[18]. Based on this construct, all mutants were generated using a QuikChange Lightning Site-Directed Mutagenesis Kit (Agilent Technologies) using primers listed in Supplementary Tab. 1 (Eurofins Genomics). All constructs were sequenced by a facility at The Centre for Applied Genomics, Sick Kids Hospital, Toronto, Canada with the AOX1 primer pair of PF$_{AOX1}$ and PR$_{AOX1}$. Freshly prepared competent cells of strain *Pichia Pastoris* SMD 1163 (*Δhis4 Δpep4 Δprb1*, Invitrogen) were electro-transformed with *Pme*I-HF (New England Biolabs) linearized plasmids containing different mutant genes using a Gene Pulser II (Bio-Rad). High-copy clone selection was performed using an in-house protocol[42,43]. In brief, a gradient of antibiotics G418 concentrations (1 mg/mL, 4 mg/mL, and 6 mg/mL) was used for the high-copy construct screening. Five colonies grown on 6 mg/mL YPD plates were then used for high-yield construct screening by an immunoblotting assay with both anti-FLAG and anti-Poly-his for further large-scale expressions. Of note, all antibodies used in this study have been well validated by the manufactures described in the specific data sheets.

### Receptor expression, purification, and labeling
The screened WT* and mutants R291A, R293A, R291AR293A, and H230A were pre-cultured on YPD [1% (w/v) yeast extract, 2% (w/v) peptone and 2% (w/v) glucose] plates containing 0.1 mg/mL G418. A single colony for each construct was inoculated into 4 mL YPD medium and cultured at 30 °C for 12 h, then transferred into 200 mL BMGY medium [1% (w/v) yeast extract, 2% (w/v) peptone, 1.34% (w/v) YNB (yeast nitrogen base) without amino acids, 0.00004% (w/v) biotin, 1% (w/v) glycerol, 0.1 M PB (phosphate buffer) at pH 6.5] and cultured at 30 °C for another 30 h. The cells were then transferred into 1 L of BMMY medium [1% (w/v) yeast extract, 2% (w/v) peptone, 1.34% (w/v) YNB without amino acids, 0.00004% (w/v) biotin, 0.5% (w/v) methanol, 0.1 M phosphate buffer at pH 6.5, 0.04% (w/v) histidine and 3% (v/v) DMSO, 10 mM theophylline] at 20 °C. 0.5% (v/v) methanol was added every 12 h. 60 h after induction by methanol, cells were harvested for purification.

The cell pellets were collected by centrifugation at 4000 × *g* for 20 min and washed one time with washing buffer (50 mM HEPES, 10% glycerol, pH 7.4) before addition of breaking buffer (50 mM HEPES, pH 7.4, 100 mM NaCl, 2.5 mM EDTA, 10% glycerol) in a ratio of 4:1 (buffer: cells). The resuspended cell pellets were subject to disruption 3 times

using a Microfluidizer at a pressure of 20,000 psi. Intact cells and cell debris were separated by low-speed centrifugation ($8000 \times g$) for 30 min. The supernatant was collected and centrifuged at $100,000 \times g$ for 2 h, and the precipitated cell membrane was then immediately dissolved in membrane lysis buffer (50 mM HEPES, pH 7.4, 100 mM NaCl, 1% MNG-3 (Lauryl Maltose Neopentyl Glycol) and 0.2% CHS (cholesteryl hemisuccinate)) with rotation 2 h or overnight at 4 °C until the membrane was dissolved. Subsequently, Talon resin (Clontech) was added to the solubilized membranes and incubated for at least 2 h or overnight under gentle agitation.

The $A_{2A}R$-bound Talon resin was washed twice with a buffer of 50 mM HEPES, pH 7.4, 100 mM NaCl, 0.02% MNG-3 and 0.01% CHS and resuspended in the same buffer. The $A_{2A}R$-bound Talon resin was then resuspended in buffer made of 50 mM HEPES, pH 7.4, 100 mM NaCl, 0.02% MNG-3 and 0.01% CHS, and combined with 10–20 fold excess of the NMR label (2-bromo-$N$-(4-(trifluoromethyl)phenyl)acetamide, BTFMA, Apollo Scientific, Stockport, UK)[4,5] under gentle agitation overnight at 4 °C. Another aliquot of NMR label was then added and incubated for an additional 6 h to ensure complete labeling. The $A_{2A}R$-bound Talon resin was washed in a disposable column extensively with buffer containing 50 mM HEPES, pH 7.4, 100 mM NaCl, 0.02% MNG-3 and 0.01% CHS, and apo $A_{2A}R$ was then eluted from the Talon resin with 50 mM HEPES, pH 7.4, 100 mM NaCl, 0.02% MNG-3 and 0.01% CHS, 250 mM imidazole and concentrated to a volume of 5 mL. The XAC-agarose gel and $A_{2A}R$ were then incubated together for 2 h under gentle agitation. The functional $A_{2A}R$ was eluted with 50 mM HEPES, pH 7.4, 0.02% MNG-3, 0.01% CHS, 100 mM NaCl, 20 mM theophylline. The eluted samples were concentrated to 1 mL by centrifugal filtration (MWCO, 3.5 KDa), and an extensive dialysis was performed to remove the XAC in the sample. The functional apo $A_{2A}R$ was then prepared for NMR. All receptors described in this manuscript were purified using poly-his resin followed with a ligand-column, in which the $A_{2A}R$ antagonist xanthine amine congener (XAC) was conjugated to Affi-Gel 10 activated affinity media.

### $^{19}$F NMR experiments
NMR samples typically consisted of 280–300 μL volumes with 20–50 μM $A_{2A}R$ in 50 mM HEPES buffer and 100 mM NaCl, doped with 10% $D_2O$. The receptor was stabilized in 0.02% MNG-3 and 0.01% CHS. All $^{19}$F NMR experiments were performed on a 600 MHz Varian Inova spectrometer using a $^{19}$F dedicated resonance probe. Typical experimental setup included a 16 μs 90° excitation pulse, an acquisition time of 200 ms, a spectral width of 15 kHz, and a repetition time of 1 s. Most spectra were acquired with 15,000–50,000 scans. Processing typically involved zero filling, and exponential apodization equivalent to 15 Hz line broadening. All NMR spectra were processed using the software collection VnmrJ 4.2 and analyzed using the program MestReNova 14.2.

### T2 measurements
$^{19}$F-labeled apo $A_{2A}R$-V229C in the buffer as described above was used for measurements of transverse relaxation time ($T_2$) by a CPMG $T_2$ pulse sequence, using a refocusing period of 133 μs, with a total transverse magnetization evolution time of 0.4, 0.8, 1.2, 1.6, 2.0, 2.4, 2.8, and 3.2 ms. The T2 values were then fitted out using the MestreNova and linewidth will be calculated based on the formulas: linewidth (Hz)=$1/\pi T_2$, which was used for the comparison of the deconvoluted linewidths of each resonance to determine 5-state model, as shown in Supplementary Fig. 3.

### Radioligand binding assay
The ligand affinities to constructs used in this study were evaluated using a saturation radioligand binding assay. Here, the full agonist [$^3$H] CGS21680 was used as a hot ligand, while the non-radio compound CGS21680 serves as cold ligand. 5 μL aliquots of purified receptor was incubated in a total volume of 50 μL assay buffer (50 mM HEPES at pH 7.4, 100 mM NaCl) with different concentrations of [$^3$H] CGS21680 at 20 °C for 120 min. Nonspecific binding was removed by saturating with 10 μM cold CGS21680. Incubation was terminated by rapid filtration performed on Whatman GF/C filter in a Millipore XX2702550 12 Position Vacuum Filtration Sampling Manifold and washed with buffer (50 mM HEPES at pH 7.4, 100 mM NaCl). The filter-bound radioactivity was determined by LS 6500 Multi-Purpose Scintillation Counter. A minimum of three independent experiments were performed, and the values were pooled to generate the mean curves prepared by GraphPad Prism® 9.

### GTPase hydrolysis assay
The GTPase hydrolysis assay was analyzed using a modified protocol of the GTPase-Glo$^{TM}$ assay (Promega)[44]. The reaction was started by mixing 300 nM $G\alpha_s\beta\gamma$ with the purified receptors in varying concentrations with a final volume of 10 μL in the buffer containing 50 mM HEPES, pH 7.4, 100 mM NaCl, 0.001% CHS, 0.05% L-MNG-3. After 30 min incubation at room temperature, 10 μL 2xGTP-GAP solution containing 10 μM GTP, 1 mM DTT and the cognate GAP was added to each well, followed with a 120 min incubation at room temperature. Then, 20 μL reconstituted GTPase-Glo$^{TM}$ reagent containing 5 μM ADP was added to each sample and incubated for another 30 min at room temperature with shaking. Luminescence was measured following the addition of 40 μL detection reagent and incubation for 10 min at room temperature using a BioTEK-Flx800 plate reader at $528 \pm 20$ nm. The relative light unit (RLU) was normalized by the values of the buffer alone. Analysis of data was performed by GraphPad Prism® 9.

### Structure-based alignments of the cation-π interaction between TM6 and TM7/H8
Amino acid conservation was evaluated by GPCRdb sequence alignment[33], which uses helical lengths and reference positions for generic residue numbering defined by manual annotation of crystal/cryo-EM structures. Receptors were selected by family. Corresponding residue positions in each class were indexed with the structure based on defined GPCRdb generic residue numbering system. TM6 region 30–35, TM7 region 54–56, and H8 region 47–50 were selected as alignment segments. Aligned sequences from family A to T were analyzed by Microsoft Excel. Statistical analyses were made by GraphPad Prism® 9.

### MD simulations
Initial conformations of the human $A_{2A}R$ were taken from crystal structures obtained with bound agonist adenosine, partial agonist LUF5833, or inverse agonist/antagonist ZM241385 (PDB IDs 2YDO[45], 7ARO[46], and 4EIY[47], respectively). The model based on PDB: 7ARO used chain A. All crystallographic non-protein atoms, including orthosteric-site ligands, were removed. To generate a consensus sequence, missing N-terminal receptor residues $M_1PIMG_5$ were omitted, as were C-terminal residues from L308 to S412. Residues H306 and V307, not resolved in PDB: 7ARO were modeled as a helical extension to helix 8 by copying them from PDB: 2YDO after alignment on backbone atoms (N, $C_\alpha$, C, O) of the shared C-terminal helical region from R300-S305. Chimeric protein inserts in intracellular loop 3 (ICL3) were removed. Missing residues in ICL3 were added with the program Loopy[48,49]. Missing side chain atoms and side chain reversions to wild-type (WT) were modeled with the program SCWRL4[50]. Disulfide bonds (C71-C159, C74-C146, C77-C166, and C259-C262) and all hydrogen atoms were placed with the GROMACS tool pdb2gmx[51]. The resulting 302-residue receptor sequence corresponds to residues S6-V307 of the human $A_{2A}R$ (Uniprot ID: P29274). Unphysical N- and C-terminal backbones were represented with cationic $NH_3^+$ and anionic $COO^-$ groups, respectively. Each structure was oriented using the Orientations of Proteins in Membranes (OPM) database web server[52] and embedded in

a hydrated micelle of 90 lauryl maltose neopentyl glycol (MNG-3) lipids using the CHARMM-GUI[53] micelle builder[54]. Detergent micelles (as opposed to lipid bilayers) were selected for consistency with experimental data. Taking the molecular weights of $A_{2A}R$ and MNG-3 to be 45 and 1 kDa, respectively, we reasoned that 90 MNG-3 detergents per micelle would approximate the 100–140 kDa apparent molecular weight of MNG-3/CHS reconstituted $A_{2A}R$ measured previously by native PAGE[18]. This composition is similar to the 96 MNG-3 detergents per micelle used in a recent simulation study of the $A_{2A}R$ and $\beta_2AR$[55]. Each system was neutralized with KCl, which was added at an excess concentration of 100 mM. For each of the three aforementioned structures, the above procedure was repeated three times, each using a different model of missing receptor atoms inside chains and ICL3. No water or ions were initially placed in the receptor's core. The cubic unit cells of these initial systems were $12.3 \pm 0.1$ nm on edge. Each of these nine WT systems was used to construct a separate R291A system and a separate R293A system (in each case by removing arginine atoms distal to the side chain $C_\gamma$ atom and converting the relevant $C_\gamma$ atom to a $C_\beta$-bonded hydrogen atom), yielding a total of 27 simulation systems.

All MD simulations were performed using GROMACS 2022.1[56]. The protein was described using the CHARMM36m force field with CMAP corrections[57]. MNG-3 parameters were obtained from the CHARMM-GUI[53] micelle builder[54]. The water model was TIP3P[58] with CHARMM modifications[59]. Water molecules were kept rigid with SETTLE[60] while other covalent bond lengths involving hydrogen were constrained with P-LINCS[61] (maximum order of 6). Lennard-Jones (LJ) nonbond interactions were evaluated using an atom-based cutoff with forces switched smoothly to zero between 1.0 and 1.2 nm. Coulomb interactions were calculated using the smooth particle-mesh Ewald method[62,63] with Fourier grid spacing of 0.12 nm and fourth order interpolation. Simulations in the isothermal-isobaric ensemble used isotropic coupling to a Berendsen barostat at 1.01325 bar with a compressibility of $4.5 \times 10^{-5}$/bar and a coupling constant of 4 ps; temperature-coupling was achieved using velocity Langevin dynamics at 310 K with a coupling constant of 1 ps. The integration time step was 2 fs. Non-bonded neighbor-lists were built to 1.23 nm and updated every 25 integration steps.

To relax detergents, water, and ions without dramatically perturbing the initial configuration of the protein and its bound ligand, the following protocol was applied to each system. Harmonic position restraints (force constant 1000 kJ/mol/nm²) were applied to all non-hydrogen atoms in the protein. After 500 steps of steepest-descent energy minimization, each system was simulated for 20 ns. Three 10-ns simulations were then conducted with position restraints on a) protein backbone atoms (N, $C_\alpha$, C, O) and all ligand non-hydrogen atoms, b) protein $C_\alpha$ atoms, and, finally c) weaker (force constant 100 kJ/mol/nm²) position restraints on protein $C_\alpha$ atoms only. Each system was then simulated without restraints for 1 μs.

### LUF5834 docking

The MD structure of LUF5833-$A_{2A}R$-Gs protein complex was used for the docking study. The side chains of T88, R107, K122, N154, L202, L235, V239, and S277 for all thermos-stabilizing Ala mutants were added using SCREAM (Side Chain Rotamer Excitation Analysis Method)[64]. The starting conformation of LUF5834 was prepared from the LUF5833-bound receptor conformation using the Maestro software[65]. We used the DarwinDock complete sampling method to predict ligand binding sites[66]. This procedure begins with an analysis of the likely binding region after replacing the 6 hydrophobic residues (I, L, V, F, Y, and W) with A to provide space for ligand docking. We then use DOCK4.0 to generate ~50,000 poses (but without energy calculations) sufficient to span the putative binding regions. To ensure complete sampling, these poses are generated in increments of 5000 and clustered into Voronoi families based on root mean square

deviation (RMSD) until the number of new families increases by <2% per increment. The binding energies of the family heads are obtained using the Dreiding force field and the top 10% by total energy are selected. Subsequently, we predicted the binding energy for all members of these top 10% families and selected the lowest-energy 100 poses for further optimization. At this point we de-alanized to restore the WT sequence using SCREAM[64], followed by full geometry energy minimization to provide independently optimized side chains for each of the 100 poses. The final docked structure with the best binding energy was selected for further analysis. As a control, we note that applying this procedure to the antagonist JDTic ligand in the X-ray structure leads to a structure that deviates by only 0.23 Å RMSD from the X-ray coordinates[67]. Based on these predictions we identified candidates for experimental validation. We selected the best biding poses based on two scoring criteria: 1) the unified cavity energy which considers the interactions of the best 100 poses with the union of all residues involved in their separate binding sites (providing a uniform comparison), 2) the snap binding energy considering all ligand-protein interactions.

### Data representation

All statistical tests such as radioligand binding assay and GTP hydrolysis assessments were conducted using GraphPad Prism 9.0. The central point of all data points gives the mean value with SD for all data unless otherwise specified.

### Reporting summary

Further information on research design is available in the Nature Portfolio Reporting Summary linked to this article.

### Data availability

The data that support this study are available from the corresponding authors upon reasonable request. The publicly available datasets with PDB accession codes 2LNL, 2R4S, 2RH1, 2YDO, 3EML, 3PWH, 4AMJ, 4EIY, 4GPO, 4L6R, 6D26, 6DGD, 6W2Y, 7ARO were used in this study for figure preparation and data analyses. Source data underlying figures are provided as a Source Data file. Source data are provided with this paper.

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

## Acknowledgements

This work was supported by the USF startup funds and Internal USF grants (L.Y.), NIH R01HL155532 (W.A.G.), and U.S. Department of Energy (DOE) Laboratory Directed Research and Development funds (C.N.). We thank Amirhossein Mafi for kindly providing MD trajectories of LUF5833-$A_{2A}$R-G$\alpha_s\beta\gamma$ for partial agonist LUF5834 docking. We thank Dr. Brian Kobilka and Dr. Jun Xu from Stanford University for providing valuable comments on the manuscript and sending us heterotrimeric G$\alpha_s\beta\gamma$. We also thank Hiran Malinda Lamabadu Warnakulasurya in assistance of receptor cell culture. Our computations used resources provided by the LANL Institutional Computing Program, which is supported by the U.S. DOE National Nuclear Security Administration under contract DE-AC52-06NA25396, and the Materials and Process Simulation Center (MSC), California Institute of Technology.

## Author contributions

L.Y. conceived and designed the research. X.W. performed the molecular biology work, generated high yield transformants, optimized receptor expression and purification, performed informatic analyses on amino acid sequence conservation and structural similarity, conducted NMR experiments and processed part of NMR data, and conducted radioligand binding assays and SDS-PAGE. C.N. performed MD simulations and bioinformatic analyses. S.K.K. performed docking experiments of LUF5834-A2AR-G$\alpha$s$\beta\gamma$. W.A.G. supervised simulation and docking experiments. L.Y. performed NMR and analyzed spectroscopy data. S.K.K., C.N., X.W., and L.Y. prepared the manuscript. All authors read and revised the manuscript. L.Y. supervised the whole project.

## Competing interests

The authors declare no competing interests.
