## [Peer Review File · Nature Communications]

Intermediate-state-trapped Mutants Pinpoint G Protein-coupled Receptor Conformational AllosteryReviewers' Comments:

Reviewer #1:

Remarks to the Author:

This manuscript by Wang et al. reports on the investigation of A2AR mutants to help to better describe the activation pathway(s) for this receptor. This study is mainly based on ^{19}F 1D NMR spectroscopy associated with *in silico* simulations. Central to this work are point mutations aim at stabilizing intermediate states to better delineate the conformational ensemble. Along with these experimental observations, the additional data from MD simulations suggests a structurally conserved and strategic cation- π interaction between transmembrane helix VI (TM6) and VII (TM7) and the additional helix VIII (H8) that would manage the opening to the cytoplasmic cavity to enable the interaction with the α G sub-unit.

As sum up in the introduction of their paper, there is a need for catching either lowly populated and/or transient sub-states on the activation of these complex dynamics and allosteric proteins. That would increase the likelihood of novel findings emerging that are of relevance to a wide range of GPCR biologists. The central hypothesis tested in this study, and what looks like to be new in the field, is to propose that residue H230 (TM6) can be "sandwiched" or not (depending of the conformational sub-state) between two arginines residues, i.e. R291 (TM7) and R293 (H8) in conjunction with a broken - or not- well-conserved ionic lock DR(TM3)Y-E(TM6) in the GPCR superfamily. Even though biochemistry and biophysics seem to be conducted well, there are some issues that dampen somehow my enthusiasm and, in particular, I still have some concerns about the connection one can established between 1D fluorine NMR data and the sequential of events in the activation of A2AR proposed here. Overall I think, based on 1D ^{19}F NMR that this model is somehow overstated, and despite the help of MD simulations, this seems to me not enough to ascertain in which proportion that model is right or wrong. We have to keep in mind that this is still quite challenging to address energy landscapes so such dynamics receptors with only one probe, and this probe is even not localized in the transmembrane part of the receptor as only solvent-exposed residues can be labeled as the chemical reaction occurs in an aqueous context. Regarding previous published investigation of the A2AR by NMR, maybe the use of amino acid residue-selective isotope labeling in other domains could help to sample more precisely the conformational ensemble and some kinetics events in this receptor associated with the activation process. For instance, observation of ^{19}F NMR signals located to some others residues (in this paper not cited in the present manuscript: Susac et al (2018) PNAS 115, 12735) gives rise to distinct distributions than those depicted here, even for a residue located quite close (position 225) than the one used here.

The argument at the heart of this study is that thanks to key mutations at the TM6 and TM7/H8 interface, intermediate states could be stabilized. And the authors claim, e.g. page 6 lines 98-100, that this represents "a substantial progress because it enables the study of discrete states along the activation pathway". In addition, this would avoid the need for T2-based spectral deconvolution. However, based on previous work made by the principal investigator (ref. 18) or more recently by Scott Prosser & colleagues (ref. 19) using ^{19}F NMR, this is not so obvious that the strategy depicted here substantially improved the reading of the A2AR conformational landscape. And I think that a need for T2-based spectral deconvolution with such low-resolved NMR spectra should be mandatory to get I) a neat evaluation of populations and II) tell more about some putative chemical exchanges that would connect these different sub-states in order to support the model of activation presented in this study. Indeed, even if this nucleus still represents a valuable alternative, in particular to investigate some kinetic barriers in GPCRs (e.g., [Manglik et al. (2016) Structural Insights into the Dynamic Process of β 2-Adrenergic Receptor Signaling. Cell 161, 1101]), the efficient ^{19}F transversal relaxation due to the large chemical shift anisotropy -which becomes the major source of line width at high magnetic fields- leads to overlaid signals. And I notice also that there is no considerations about variations in chemical shift differences between the inactive and active regions in these 1D ^{19}F spectra along the different mutants used. Could 2D ^{19}F exchange spectroscopy or saturation transfer NMR experiments help to characterize some chemical exchanges [e.g. Eddy et al. (2018) Cell, 172,

68; Susac et al (2018) PNAS 115, 12735]?

Related to my previous comment, would it be beneficial to improve the spectral resolution to work with partially deuterated WT and mutant A2AR as it has been proven that yeast is compatible with a growth in D₂O? I have no background if this would be beneficial to the balance between dipolar versus CSA relaxation mechanisms in the case of fluorine nucleus.

Importantly, through the manuscript this is difficult to catch how they decided to mutate R291 and R293 residues. The argumentation developed here does not directly concern these residues.

For me this is an important point: the relevancy of studying the conformational landscape of A2AR in a detergent solution is questionable especially since the publication of a similar approach but with A2AR embedded in a lipid nanodisc (ref. 19). The conformational landscape, but also chemical exchanges, look quite different between A2AR/detergent vs. A2AR/lipid nanodiscs. For instance, this is quite surprising in the apo state to observe that when associated with detergent micelles, the inactive states of the apo receptor are poorly populated compared to the active ones. And we observe an opposite trend with the receptor in lipid disc which seems to me more logical, i.e. to sample various inactive states at the entrance of the activation process instead for a ligand to directly bind to a pre-active-like state. I think this is a point that should be discussed somewhere in the text. Moreover, in Susac et al (2018) PNAS 115, 12735, in the apo state, the inactive state(s) is(are) also mostly populated based on ¹⁹F NMR of A2AR in a detergent solution.

Overall, the data displays here should be put more in a perspective with the literature. For instance, while the inactive states S1 and S2 with a same sample and in the same conditions (same detergent and ¹⁹F Larmor frequency; ref. 18) seems to display a fast chemical exchange between them at the ¹⁹F NMR chemical shift timescale while it is not in the present study (page 7, Figure 1) ?

Additional comments:

- the authors added once G(αS/β/γ), i.e. in the presence of full agonist NECA in the presence of WT receptor. Would it be useless to add a G protein with the three distinct mutants in the apo or holo states as well?

- the use of H230A mutant is interesting, but why NMR experiments were conducted in the apo state only?

- about the link between aqueous exposure and chemical shift (but also linewidth) of the ¹⁹F probe, the seminal paper of the fluorotyrosine alkaline phosphatase is often cited (though not in the present manuscript) (Sykes, 1974, PNAS, 71, 469). Maybe it would be informative to the reader to cite some papers to argue that the S1-to-S5 progression of the ¹⁹F NMR chemical shift from the low to high magnetic field is due to an increase of to aqueous exposure. As, the fully activated state is stated to be S5 (highest magnetic field; equivalent to A3 in ref. 19) in the present manuscript while it is S4 (=A2) in ref. 19, which means not the the most aqueous exposed if we follow that logic.

- page 7, lines 113-115. About the S1 sub-state, it is written that LUF5834 does not perturb S1 with the mutant R291A compared to R293A and R291AR293A. But the s/n ratio between the different samples in Figure 1 are not the same (R291A mutant displays the best s/n compared to the other mutants so maybe it could help to observing S1 (?)).

- Figure 1, page 7. In the apo form and R293A mutant, it seems there is a peak between S3 and S4 (?).

- page 7, lines 118-121: regarding the lack of resolution between some of the active sub-states, this

is not so obvious that S4 sub-state is not present with the R293A mutant. And again, regarding S1, the large 19F chemical shift anisotropy associated with putative unfavorable chemical exchanges become the major sources of line width, thus leading to a decreased signal-to-noise ratio (and reduced spectral resolution). Based on these 1D NMR spectra in Figure 1, I would not bet a dime on such interpretation.

- page 8, lines 126-130. It seems to me, but I maybe wrong, that the concept that the bundle packing between TM6 and TM7/TM8 is critical in maintaining the inactive conformational states S1 and S2 is not entirely novel. If not, maybe some references to the literature would be beneficial to the readers.

- As R291A seems to catch almost the S4 sub-state only without the need of any thermostabilized mutations, perhaps this would ease some X-ray and/or cryoEM structural investigations to reinforce the model of activation proposed herein?

- page 10, lines 146-147. I would suggest to add GTP-tp-G protein binding experiments catalyzed in the presence of these different mutants compared to the WT.

- page 11, line 157-161. I think these two sentences should be moved elsewhere, probably in the methods section.

- top of page 16. some considerations on H230/R293 interactions but nothing concerning R291. I think some comment should be added related to Figure 8.

- Overall, Figure legends are not descriptive enough on my point of view.

Reviewer #2:

Remarks to the Author:

This manuscript provides another piece to the puzzle of structural changes in GPCRs. There is a mixture of computational and wetlab work but most of the manuscript's focus is on the experimental work. Hidden in the methods is that this work is based on detergent solubilized GPCRs and not in the native cell membrane environment.

General Comments:

1. Lack of Cell Membrane: The NMR work presented here has the GPCRs in detergent which is far from the native state. How much can one believe this study compared to a GPCR that is placed in a more native-like state of a bilayer. Detergents and micelles might stabilize certain structures of the protein core, but details of functional mechanism might be perturbed. There is no discussion of this assumption in the manuscript and the proteins environment is only briefly alluded to if you read the methods.

2. Simulation Results: The simulation aspect of this manuscript is quite limited. There is single extended data figure 4 that shows how certain mutants behave differently from the WT. The description of these results should be providing the reader with a bit more detail beyond the results being consistent with ion lock breakage and helix 6 rotation. Please guide the reader the quantitative results that show this.

3. Simulation setup: As with the wetlab work, the authors decide to simulate a micelle. This is disappointing in that it would be nice to see if there is an effect to structure/function in the more native membrane environment. Moreover, the selection of micelle size is not clear. Why were 90 MNG-3 lipids decided to represent the micelle? Is the micelle that covers the GPCR in experiment this size

or different? Is this micelle spherical or asymmetrical and by how much? How does the lipid headgroup interact with the cyto/extracellular parts of the GPCR? Could this alter structure and function?

POINT-TO-POINT RESPONSES TO REVIEWERS' COMMENTS

We thank the Reviewers for their helpful comments and for the opportunity to improve our manuscript. Each criticism is addressed below.

Response to Reviewer #1:

This manuscript by Wang et al. reports on the investigation of A2AR mutants to help to better describe the activation pathway(s) for this receptor. This study is mainly based on ¹⁹F 1D NMR spectroscopy associated with in silico simulations. Central to this work are point mutations aimed at stabilizing intermediate states to better delineate the conformational ensemble. Along with these experimental observations, the additional data from MD simulations suggests a structurally conserved and strategic cation- π interaction between transmembrane helix VI (TM6) and VII (TM7) and the additional helix VIII (H8) that would manage the opening to the cytoplasmic cavity to enable the interaction with the α G sub-unit.

As a sum up in the introduction of their paper, there is a need for catching either lowly populated and/or transient sub-states on the activation of these complex dynamics and allosteric proteins. That would increase the likelihood of novel findings emerging that are of relevance to a wide range of GPCR biologists. The central hypothesis tested in this study, and what looks like to be new in the field, is to propose that residue H230 (TM6) can be “sandwiched” or not (depending of the conformational sub-state) between two arginine residues, i.e. R291 (TM7) and R293 (H8) in conjunction with a broken -or not- well-conserved ionic lock DR(TM3)Y-E(TM6) in the GPCR superfamily. Even though biochemistry and biophysics seem to be conducted well, there are some issues that dampen somehow my enthusiasm and, in particular, I still have some concerns about the connection one can establish between 1D fluorine NMR data and the sequential of events in the activation of A2AR proposed here. Overall, I think, based on 1D ¹⁹F NMR that this model is somehow overstated, and despite the help of MD simulations, this seems to me not enough to ascertain in which proportion that model is right or wrong.

-We thank the Reviewer for constructive and insightful comments regarding the receptor activation process. The activation process proposed in this manuscript in part builds upon our previous study ¹, in which an ionic lock DR102Y^{TM3}-E228^{TM6} switch/exchange between two inactive states (S1 and S2) was identified using the NMR relaxation experiments. Of note, the ionic lock switch between the S1 and S2 was also observed in the β_2 AR ², along with their conformational structures in different receptors, shown in Extended Data Fig.2 in the manuscript.

In this manuscript, we identify an additional conserved cation- π interaction/lock R291^{TM7}/R293^{H8}-H230^{TM6} that plays a role in switching the inactive states (S1-2) to the active states (S3-5) through the separation of TM6 domain from TM7/H8, resulting in the opening of G protein-binding cavity. In particular, we determined that R291 and R293 play respective roles in regulating S1 and S2 states through sandwiching the H230 residue. Importantly, our manuscript identifies an intermediate state (S4) that was superimposed with other states in previous works ^{1,3,4}. This enabled us to extend the existing 4-state model ¹ to a 5-state model, and to predominantly populate this intermediate state for direct assessment. As advised by the reviewer, we have made a slight change in our model to emphasize the role of cation- π switch in conformational equilibrium shifting from the inactive states (S1-2) to the active states (S3-5) in Fig. 5 of the revised version. This is better aligned with the data presented in our manuscript and avoids overstatement.

In response to the Reviewer's concern about our use of the word "sequential", we have removed that word from the manuscript title, which is now "**Intermediate-state-trapped Mutants Pinpoint G Protein-coupled Receptor Conformational Allostery**".

We have to keep in mind that this is still quite challenging to address energy landscapes so such dynamics receptors with only one probe, and this probe is even not localized in the transmembrane part of the receptor as only solvent-exposed residues can be labeled as the chemical reaction occurs in an aqueous context. Regarding previous published investigation of the A2AR by NMR, maybe the use of amino acid residue-selective isotope labeling in other domains could help to more precisely sample the conformational ensemble and some kinetics events in this receptor associated with the activation process. For instance, observation of ^{19}F NMR signals located to some others residues (in this paper not cited in the present manuscript: Susac et al (2018) PNAS 115, 12735) gives rise to distinct distributions than those depicted here, even for a residue located quite close (position 225) than the one used here.

-As mentioned by the Reviewer, it is challenging to map the conformational energy landscape of a receptor. For GPCRs, and especially for family A GPCRs, the TM6 domain is regarded as the most sensitive subdomain in responding to receptor activation. This implies that a probe on the TM6 would undergo the largest conformational change (and give the highest conformational resolution) compared to other regions. We previously evaluated another site A289C on the TM7 domain during the screening process^{5,6}, but this showed less conformational resolution than the TM6 labeling. Therefore, we used the labeling of V229C on TM6 for this research.

As mentioned by the reviewer, we would like to point out that the A289C site (It presented a better conformational resolution than the 225 site) in the publication of (Susac et al (2018) PNAS 115, 12735) was not quite comparable to our study because of the system difference. We addressed part of reasons in our recent review to **Structure**, titled " ^{19}F NMR: A promising tool for dynamic conformational studies of G protein-coupled receptors". This includes:

(1) two different ^{19}F probes (TET and BTFMA) were used - these two probes have presented significant differences in both $\text{A}_{2\text{A}}\text{R}$ and $\beta_2\text{AR}$ GPCR studies^{2,7-9}, For instance, the activation of the receptor in BTFMA labeled system shifted the conformational equilibrium to high magnetic field, whereas in the TET system the conformational equilibrium shifted to downfield;

(2) two different labeling strategies were used - in their research an in-membrane labeling was applied but there was no information regarding the effect of this pre-labeling on the receptor reconstitution and thus the conformational profiles;

(3) two different detergent reconstitution systems were used - in our study we used the MNG-3/CHS system while they used the DDM/CHS system¹⁰, and these two systems had shown ^{19}F spectral differences, including conformational resolution, for the $\beta_2\text{AR}$ studies^{2,11};

(4) two different expression systems were used as well - in our study we used yeast while they used the insect cell¹⁰;

(5) two different purification procedures were used - in our study a ligand-column purification step was compiled to remove non-functional receptors while there is no report for this step in their study, and we do not know consequence of these changes.

In brief, in our system the site V229C exhibited a much better conformational resolution than the site of A289C. To highlight these topics in the manuscript, we added the following text on pages 22-23, lines 288 - 295:

^{19}F represents one of the most sensitive nuclei to microenvironmental changes and it has been utilized to profile the conformational states of GPCRs for many years¹². However, due to the large chemical shift anisotropy effect (CSA) of macromolecular ^{19}F NMR, conformational states often

suffer from spectral overlap resulting from linewidth broadening. This has historically occurred with both BTFMA, and TET labeled A_{2A}R^{1,4,8}. Furthermore, differences in chemical probes, sample preparation procedures, and solubilizing amphiphiles can affect the resulting ¹⁹F spectra of A_{2A}R⁷ and β₂AR^{2,4,9}.

For the question regarding only an aqueous residue was labeled. We appreciate the Reviewer's constructive and insightful suggestions. It was a great idea to examine the conformational transitions and dynamics in the middle of the receptor, including transmembrane domains. However, we also note that the application of ¹⁹F NMR in probing the GPCR conformational states in the transmembrane domains is still extremely challenging. Indeed, a tri-fluorinated amino acid incorporation was recently explored for the CB1¹³ receptor, which was also incorporated into a solvent exposure area. There could be other ways like using fluorinated amino acid (such as fluorinated Tyr, Phe, or Trp) to biosynthetically culture cell for protein production, but multiple sites are always incorporated unless an extensive site-directed mutagenesis was performed, along with a low expression of GPCRs, making this task difficult to pursue in GPCRs for the study of conformational transition and dynamics.

We would like to emphasize that our lab is making every effort towards such developing methods, but much work remains to establish a mature system close to chemical conjugation in probing the conformational states. This is due to various reasons, including a significant production decrease of receptors when unnatural amino acid genetic incorporation is applied. These studies are even more challenging because a large quantity of GPCR is required for NMR experiments. This becomes even worse in macromolecular ¹⁹F NMR because of the large CSA effects of the ¹⁹F nucleus. We hope to make progress in this area over the next few years.

The argument at the heart of this study is that thanks to key mutations at the TM6 and TM7/H8 interface, intermediate states could be stabilized. And the authors claim, e.g. page 6 lines 98-100, that this represents “a substantial progress because it enables the study of discrete states along the activation pathway”. In addition, this would avoid the need for T2-based spectral deconvolution. However, based on previous work made by the principal investigator (ref. 18) or more recently by Scott Prosser & colleagues (ref. 19) using ¹⁹F NMR, this is not so obvious that the strategy depicted here substantially improved the reading of the A_{2A}R conformational landscape. And I think that a need for T2-based spectral deconvolution with such low-resolved NMR spectra should be mandatory to get I) a neat evaluation of populations and II) tell more about some putative chemical exchanges that would connect these different sub-states in order to support the model of activation presented in this study. Indeed, even if this nucleus still represents a valuable alternative, in particular to investigate some kinetic barriers in GPCRs (e.g., [Manglik et al. (2016) Structural Insights into the Dynamic Process of β₂-Adrenergic Receptor Signaling. Cell 161, 1101]), the efficient ¹⁹F transversal relaxation due to the large chemical shift anisotropy -which becomes the major source of line width at high magnetic fields- leads to overlaid signals. And I notice also that there is no considerations about variations in chemical shift differences between the inactive and active regions in these 1D ¹⁹F spectra along the different mutants used. Could 2D ¹⁹F exchange spectroscopy or saturation transfer NMR experiments help to characterize some chemical exchanges [e.g. Eddy et al. (2018) Cell, 172, 68; Susac et al (2018) PNAS 115, 12735]?

-We regret that we didn't articulate these topics well in the previous version of this manuscript. When we mentioned “a substantial progress”, we were referring to the intermediate state trapped mutants that allow us to study the function of each conformational state and their roles in the activation process, avoiding the interference from other states. This would be difficult to pursue in the previous study using the WT* construct because all conformational states are presented in

the construct. To clarify our meaning, we have changed the relevant sentence on page 6, lines 99 - 106 to:

Here, we mitigate this limitation with point mutations in TM7/H8 that change the profile of stability among detergent solubilized receptor states. We also reduced the detergent MNG-3 concentration to 0.02% compared to previous studies (0.1%). A lower concentration of detergent has been reported to provide a better NMR resonance resolution for dodecylmaltoside (DDM) solubilized rhodopsin¹⁴, and our present study indicates that maltose-neopentyl glycol/cholesteryl hemisuccinate (MNG-3) has a similar property to the DDM. These advances facilitate more precise characterizations of intermediate states and their roles in activation pathways.

In these conformation-biased mutants, we benefit from the elimination of a particular state such as the fully activated state S5 in the R291A mutant. Thus we can study whether the S4 state interacts with G proteins and its ability to regulate nucleotide exchange without interference from the S5 state. This type of work is ongoing in our lab currently. The data in the newly added Extended Data Fig.7 (page 15) in the revised version that the S4 state in the mutant R291A exhibits a limited GTP hydrolysis capacity, compared to WT*.

Regarding the T2 deconvolution, this research was a follow-up study of our previous work published in Nature (2016) and Nature Communications (2018). In those two studies, a T2 deconvolution was applied, but the main resonance could only be deconvoluted into two states consisting of S3 and S3'. In those two studies, the chemical shifts for each state were largely based on the fitting process using MestreNova after a T2 deconvolution of the peak containing the S3 and S3' states. In the current study, by using the intermediate-state trapped mutants we were able to delineate the same NMR resonance (the peaks 2 and 3 in the Extended Data Fig. 3) into three states (S3, S4, and S5) by decreasing 9 variances in the fitting process, including chemical shift, linewidth, and resonance intensity of each state in the previous study into 6 variances, decreasing the uncertainty of T2-based fitting as well. For instance, with T2-based conformational deconvolution, it is difficult to distinguish the superimposed resonances/states with same/similar T2 values or when the subpopulations of each delineated conformation are significantly different, especially if a low population state exists in the mixed resonance. This was one of downsides in the previous study that we only delineated the resonances into two states because the high energy state sometimes are often little populated both in our previous study in MNG-3/CHS¹ and a late research in HDL systems⁴. As shown in Fig. 2, this high energy low populated state (S5) state can be well-defined using our conformation-biased mutants. The revisited T2 measurement experiments (Extended Fig.3) in this low MNG-3 concentration system also support our 5-state model, consistent with conformation-biased mutants-led deconvolution.

Regarding the consideration of chemical shift difference between inactive and active states in different mutants, considering the complexity of spectral deconvolution with 3 variances (chemical shift, linewidth, and intensity) for each conformational component, we thus fixed the chemical shift for each conformational state with a 0.05 ppm flexibility when performed spectral deconvolution for different mutants, which could arise from conformational exchanges, as stated in the manuscript on page 10. Considering the focus of this manuscript and workload for conformational transitions and dynamics of different mutants as a function of different ligands and downstream transducers, we are working on these data and wish to be published in a separate manuscript in the future. Thanks a lot for the advice.

Of note, the T2-based measurement was also used to validate the linewidths of each conformational component, as shown in Extended Data Fig. 3, to support the deconvolutions we performed here based on conformation-biased constructs. Considering three variances for each conformational state in the deconvolution process

that lead to the complexity of spectral fitting, we defined the chemical shifts of each state using the values presented in these conformation-biased mutants with ± 0.05 ppm variations for each state during the fitting process, considering conformational dynamics in varied samples. The linewidth and intensity of each state are then determined through the best fitting of each ^{19}F NMR spectrum with a minimal fitting error, which serves as a part of standard deviation for each conformation component shown in Fig.2d.

Related to my previous comment, would it be beneficial to improve the spectral resolution to work with partially deuterated WT and mutant A2AR as it has been proven that yeast is compatible with a growth in D₂O? I have no background if this would be beneficial to the balance between dipolar versus CSA relaxation mechanisms in the case of fluorine nucleus.

-In principle, the solvent isotope effect on the chemical shift from H₂O to D₂O can be observed for ^{19}F resonances, leading to chemical shift differences up to 0.25 ppm¹⁵⁻¹⁷. Considering the cost of D₂O for 6 L cell culture and the amount of the purified receptors from 6 L just allowing us to

Fig.R1: ^{19}F NMR spectral comparison for the WT* in H₂O and D₂O.

conduct a couple of NMR experiments even using H₂O, we conducted an experiment to replace H₂O with D₂O solvent in the purification process to reconstitute A_{2A}R first, as shown in Fig. R1. Significant chemical shifts for different states were not observed, though slightly higher resolution could be observed with the D₂O sample. However, this improvement seems to be compromised with linewidth broadening. Therefore, we stuck to the original well-established system for this study. Still, the D₂O deuterated culture could bring the benefit for compensating the negative effects of CSA relaxation. We may pursue this in the future, but not now because of the budget cost and workload of re-conducting all experiments done in this manuscript and the uncertainty of whether the D₂O cultured receptor can bring the benefit of improving conformational resolution of different states.

Importantly, through the manuscript this is difficult to catch how they decided to mutate R291 and R293 residues. The argumentation developed here does not directly concern these residues.

-The mutations of R291A and R293A were primarily determined from the MD simulations with an initial expectation to eliminate the fully activated state S5, as shown in the new Fig. 1, page 5, which was the original Extended Data Fig. 3. We have rephrased the description in the manuscript to explicitly state the connection between the initial design and the subsequent findings. Relevant new text on pages 5-6, lines 78 - 91 is:

With the help of molecular dynamics (MD) simulations of detergent solubilized receptor and molecular docking, we generated models of $A_{2A}R$ - $G\alpha_s\beta\gamma$ bound to either a partial agonist (LUF5834) or a full agonist (NECA)¹⁸, representing partially activated $A_{2A}R$ - $G\alpha_s\beta\gamma$ (Fig.1b) and fully activated $A_{2A}R$ - $G\alpha_s\beta\gamma$ complexes (Fig.1c), respectively, in addition to a pre-coupled complex (Fig.1a). The NECA- $A_{2A}R$ - $G\alpha_s\beta\gamma$ model reveals two intermolecular salt bridges between intracellular loop 3 (ICL3) in the $A_{2A}R$ and the AHD of $G\alpha_s$ (Fig. 1c). Importantly, these salt bridges are lost in the predicted NECA- $A_{2A}R$ - $G\alpha_s\beta\gamma$ complex (Fig. 1b). This ligand-specific behavior inspired us to design receptor mutations that could quench the signal from the full agonist, with expectation of eliminating the corresponding conformational state and thus stabilizing intermediate states. To our surprise, the mutants based on these presumptions created a series of conformation-biased constructs, including those trapped intermediate states. These advances provided insights into the roles of R291 and R293 in the receptor activation process through allosterically modulating the opening of G protein binding cavity.

For me this is an important point: the relevancy of studying the conformational landscape of A2AR in a detergent solution is questionable especially since the publication of a similar approach but with A2AR embedded in a lipid nanodisc (ref. 19). The conformational landscape, but also chemical exchanges, look quite different between A2AR/detergent vs. A2AR/lipid nanodiscs. For instance, this is quite surprising in the apo state to observe that when associated with detergent micelles, the inactive states of the apo receptor are poorly populated compared to the active ones. And we observe an opposite trend with the receptor in lipid disc which seems to me more logical, i.e. to sample various inactive states at the entrance of the activation process instead for a ligand to directly bind to a pre-active-like state. I think this is a point that should be discussed somewhere in the text. Moreover, in Susac et al (2018) PNAS 115, 12735, in the apo state, the inactive state(s) is(are) also mostly populated based on ¹⁹F NMR of A2AR in a detergent solution.

-It could be that the HDL system is more physiologically relevant compared to micelle systems. We also agree on this under a perfect scenario. However so far only a few GPCRs, including $A_{2A}R$ and β_2AR , have been subjected to ¹⁹F NMR to profile their conformational states. Including inactive, partially activated, and fully activated states enabled us to compare the benefits of using each system. In both $A_{2A}R$ and β_2AR studies the HDL reconstituted receptors exhibited over-activated conformational profiles compared to those in the MNG-3/CHS reconstitution system as shown in Fig.R2, but the reason behind this phenomenon is still unclear^{4,19}. This also implied that the data from HDL system were not consistent with pharmacological studies, in which the β_2AR was a low constitutive receptor which should not present a large portion of activated states^{19,20}.

Though $A_{2A}R$ is relatively a high constitutive receptor, compared to β_2AR , it is not as high as shown in the HDL system probed by ¹⁹F NMR. From this perspective, the ¹⁹F NMR profiled conformational states in the MNG-3/CHS system are more aligned with pharmacological assessments while the current HDL system still needs to be improved for this type of research. This also indicates that MNG-3/CHS is a better system to study the receptor activation process

while the HDL system might be a more desirable system to study the effects of lipids on the receptor activation and to obtain structures for the fully activated complex with G proteins using the cryo-EM or x-ray. These discrepancies have also been described in our recent review paper to **Structure**, describing the progress of ^{19}F NMR study in GPCR for the last two decades.

Fig.R2: ^{19}F NMR spectral comparison for the receptors in detergent and HDL. **a** The conformational profiles probed by ^{19}F NMR for the $A_{2A}R$ receptor in detergent MNG-3/CHS and HDL systems (the figure was adapted from Ye, et al, Nature 2016; Huang, et al., Cell, 2018). **b** The conformational population distribution differences probed by ^{19}F NMR for the β_2AR in detergent MNG-3/CHS and HDL systems (the figure was adapted from Staus, et al., J. Biol. Chem, 2019).

Regarding the comments that ^{19}F NMR of $A_{2A}R$ in Susac et al (2018) populated the conformational profile differently, which could be due to the different systems used, including sample preparation, reconstitution system, labeling system, etc., we have added discussions on pages 22-23, lines 288-295 in the manuscript, along with the previous discussions in this document.

^{19}F represents one of the most sensitive nuclei to microenvironmental changes and it has been utilized to profile the conformational states of GPCRs for many years¹². However, due to the large chemical shift anisotropy effect (CSA) of macromolecular ^{19}F NMR, conformational states often suffer from spectral overlap resulting from linewidth broadening. This has historically occurred with both BTFMA, and TET labeled $A_{2A}R$ ^{1,4,8}. Furthermore, differences in chemical probes, sample preparation procedures, and solubilizing amphiphiles can affect the resulting ^{19}F spectra of $A_{2A}R$ ⁷ and β_2AR ^{2,4,9}.

For a detailed description of manuscript changes related to this Reviewer's point about detergent micelles vs. lipid nanodiscs, please see our response to general comment #1 from Reviewer #2. Also, we now more clearly state that these experiments and simulations were conducted with detergent micelles, as we describe in more detail at the top of our response to Reviewer #2.

Overall, the data displays here should be put more in a perspective with the literature. For instance, while the inactive states S1 and S2 with a same sample and in the same conditions (same

detergent and ^{19}F Larmor frequency; ref. 18) seems to display a fast chemical exchange between them at the ^{19}F NMR chemical shift timescale while it is not in the present study (page 7, Figure 1) ?

-This is a very insightful comment. Yes, the conformational resolution of the $A_{2A}R$ in this manuscript is better than the previous one, as we used a relatively low concentration of MNG-3 (0.02% MNG-3, described in the method part, page 26) in this research, compared to that used in the previous study ¹. Interestingly, this phenomenon was also observed in the detergent DDM (or DM) reconstituted rhodopsin system by Klein et al ¹⁴, as shown in the figure below. Their study also indicated that different concentrations of the detergents, such as 0.1% DDM vs 10% DDM, or 1% OG vs 10% OG, produce slightly different conformational profiles for the receptor with different chemical shift timescale and conformational dynamics. We thus added some sentences in the manuscript on page 6, lines 99 - 106 as below:

Here, we mitigate this limitation with point mutations in TM7/H8 that change the profile of stability among detergent solubilized receptor states. We also reduced the detergent MNG-3 concentration to 0.02% compared to previous studies (0.1%). A lower concentration of detergent has been reported to provide a better NMR resonance resolution for dodecylmaltoside (DDM or DM) solubilized rhodopsin¹⁴ as shown in Fig.R3 below, and our present study indicates the maltose-neopentyl glycol/cholesteryl hemisuccinate (MNG-3) has a similar property to the DDM. These advances facilitate more precise characterizations of intermediate states and their roles in activation pathways.

Fig.R3: The spectral profiles for a typical GPCR-rhodopsin in different concentrations of detergents. (The figure was adapted from Klein, et al., PNAS, 1999.)

Additional comments:

-the authors added once G(α S/ β / γ), i.e. in the presence of full agonist NECA in the presence of WT receptor. Would it be useless to add a G protein with the three distinct mutants in the apo or holo states as well?

-In this manuscript, the purpose of the addition of $G\alpha\beta\gamma$ to the NECA saturated receptor is solely to serve as a control to determine the fully activated S5 state. Considering that the manuscript is focused on conformational distinction among different mutants in response to ligand binding instead of signaling transduction that involves downstream signaling partners such as G proteins, GRKs, and β -arrestins, we would like to present those data with $G\alpha\beta\gamma$ proteins in a separate

manuscript that will study exclusively complexes with different conformational states, dynamics, and signaling outputs.

-the use of H230A mutant is interesting, but why NMR experiments were conducted in the apo state only?

-The purpose of presenting H230A was to examine if the elimination of H230 residue will quench both S1 and S2 states because it interacts with R291 which is related to the S2 state formation while interaction with R293 is related to the S1 state formation. The interruption of both interactions will result in the disappearance of both states, as confirmed in apo sample, comparable with apo sample for other mutants.

-about the link between aqueous exposure and chemical shift (but also linewidth) of the ^{19}F probe, the seminal paper of the fluorotyrosine alkaline phosphatase is often cited (though not in the present manuscript) (Sykes, 1974, PNAS, 71, 469). Maybe it would be informative to the reader to cite some papers to argue that the S1-to-S5 progression of the ^{19}F NMR chemical shift from the low to high magnetic field is due to an increase of aqueous exposure. As, the fully activated state is stated to be S5 (highest magnetic field; equivalent to A3 in ref. 19) in the present manuscript while it is S4 (=A2) in ref. 19, which means not the the most aqueous exposed if we follow that logic.

-Thank you very much for the suggestion. We now cite some papers with a paragraph included in the manuscript, please refer to lines 135-139, pages 8-9, with the sentence below:

This is consistent with the previous studies showing that protein ^{19}F NMR signals tend to shift downfield and broaden as the ^{19}F probe encounters a more hydrophobic (less aqueous) environment ^{21,22}. The populations of each conformational state can also be calculated based on the integrals of each delineated resonance, as shown in Fig.2d.

Regarding the equivalence of the S5 in ref. 19, we have commented on this in our previous review paper in the journal of **Structure**. Considering that a scaffold protein MSP was used in the HDL system, the rotation of BTFMA-labeling site of V229C as the receptor is activated could lead fluorine atoms of the BTFMA to approach another hydrophobic electrostatic cloud created by the MSP scaffold protein when the probe is leaving the hydrophobic core of G protein binding cavity. These sandwiched electrostatic clouds lead to a reversed downfield shift of BTFMA chemical shift during the activation from one hydrophobic core to the other. As a consequence, the fully activated conformational resonance in the HDL system appeared relatively downfield in an abnormal way or not a linearized activation manner. However, in the MNG-3 system, this scenario doesn't exist because the entire environment surrounding the receptor is homogenous.

-page 7, lines 113-115. About the S1 sub-state, it is written that LUF5834 does not perturb S1 with the mutant R291A compared to R293A and R291AR293A. But the s/n ratio between the different samples in Figure 1 are not the same (R291A mutant displays the best s/n compared to the other mutants so maybe it could help to observing S1 (?)).

-Yes, it is true that the S/Ns are not the same for different mutants because the expression levels are different. However, the appearance of the S1 states in different mutants can be further evaluated as in Figure 3 when different ligands were added into mutants. We conclude that the S1 state didn't appear in the apo samples of R293A and R291AR293A but a small portion could be sampled in the ZM bound R293A and LUF5834 bound R291AR293A receptors, as shown in Fig. 3b and Fig.3c.

-Figure 1, page 7. In the apo form and R293A mutant, it seems there is a peak between S3 and S4 (?).

-We performed a deconvolution for all spectra and subpopulations for each conformational state, as shown in Figure 2, so that the readers can better understand the distributions of different conformational states in different mutants. Our best fitting process didn't result in a peak between S3 and S4.

-page 7, lines 118-121: regarding the lack of resolution between some of the active sub-states, this is not so obvious that S4 sub-state is not present with the R293A mutant. And again, regarding S1, the large ¹⁹F chemical shift anisotropy associated with putative unfavorable chemical exchanges become the major sources of line width, thus leading to a decreased signal-to-noise ratio (and reduced spectral resolution). Based on these 1D NMR spectra in Figure 1, I would not bet a dime on such interpretation.

-Thanks for the comments. We assume the reviewer was talking about the R291AR293A regarding the existence of the S4 state because the S4 state was obvious in the mutant R293A. Regarding the existence of S4 substate in the R291AR293A mutant, we performed a deconvolution for all spectra, as shown in the new Fig. 2. Indeed, there is a small portion of the S4 state based on our best fitting. A sentence has been added on page 9, lines 143 - 145.

The R291AR293A double mutant allowed us to better reveal discrete states S3 and S5 as well, along with a small portion of the S4 state.

Regarding the S/N of the S1, we fully agree with the reviewer's opinion. So, we rephrased our statements for these descriptions with the support of a detailed deconvolution (pages 9-10).

-page 8, lines 126-130. It seems to me, but I maybe wrong, that the concept that the bundle packing between TM6 and TM7/TM8 is critical in maintaining the inactive conformational states S1 and S2 is not entirely novel. If not, maybe some references to the literature would be beneficial to the readers.

-Thanks. We have added some references here. Page 11, lines 175 - 176.

Combining these results, we hypothesize that bundle packing between TM6 and TM7/H8 is critical in maintaining the inactive conformational states S1 and S2, though previous studies had indicated bundle of TM6 and TM7/H8 were involved in stabilizing the inactive states²³⁻²⁵, in which two inactive states with differential H8/TM7 movements could occur^{26,27}.

-As R291A seems to catch almost the S4 sub-state only without the need of any thermostabilized mutations, perhaps this would ease some X-ray and/or cryoEM structural investigations to reinforce the model of activation proposed herein?

-Thanks for the insights. We are working on it and wish we can make some progress in the near future, but we may encounter difficulty because it is an intermediate state and no intermediate complex has been resolved so far. Also which condition of heterotrimeric G proteins should be utilized for intermediate complex structure is still in question and no previous example can be referenced. We put the sentence below in the discussion (page 23, lines 299 - 302).

These mutants will facilitate further studies of intermediate complexes in GPCR signaling, including resolving the intermediate complex structures using intermediate state trapped mutants. This would provide the additional information necessary to define receptor activation beyond the simple two-state model.

-page 10, lines 146-147. I would suggest to add GTP-tp-G protein binding experiments catalyzed in the presence of these different mutants compared to the WT.

-Thanks. We thus added GTPase hydrolysis experiments into the new version to present the differences for these different mutants, in comparison to the WT, but qualitatively connecting the

receptor conformational profiles to the signaling outputs. Please see the new Extended Data Fig. 7 and corresponding description on pages 15-16.

-page 11, lines 157-161. I think these two sentences should be moved elsewhere, probably in the methods section.

-Thanks for pointing this out. We have now moved these two sentences into the section "Receptor expression, purification, and labeling" in the method part, page 26.

-top of page 16. some considerations on H230/R293 interactions but nothing concerning R291. I think some comment should be added related to Figure 8.

-We have added new Extended Data Fig. 11 to describe H230/R291 interactions, page 20.

-Overall, Figure legends are not descriptive enough on my point of view.

-We have slightly revised the figure (Fig.5 now) and put additional descriptions. We also tried to avoid overstatements, as pointed out by the reviewer.

Response to Reviewer #2:

This manuscript provides another piece to the puzzle of structural changes in GPCRs. There is a mixture of computational and wetlab work but most of the manuscript's focus is on the experimental work. Hidden in the methods is that this work is based on detergent solubilized GPCRs and not in the native cell membrane environment.

-We regret that the system under evaluation was not sufficiently clear in our original submission. To enhance clarity, we have added the following (italicized) texts on

page 5:

With the help of molecular dynamics (MD) simulation of the detergent solubilized receptor...

Page 6:

Previously, we established a ^{19}F -NMR system for probing the $\text{A}_{2\text{A}}\text{R}$, in which two active and two active-like conformational states were delineated in detergent micelles and lipid nanodiscs using T2-based spectral deconvolution.....Here, we mitigate this limitation with point mutations in TM7/H8 that change the profile of stability among detergent solubilized receptor states...

Page 8:

Specifically, we reexamined the conformational progression from the inactive state (S1) to the fully activated state (S5) for the $\text{A}_{2\text{A}}\text{R}$ receptor in MNG-3 detergent with CHS...

General Comments:

1. Lack of Cell Membrane: The NMR work presented here has the GPCRs in detergent which is far from the native state. How much can one believe this study compared to a GPCR that is placed in a more native-like state of a bilayer. Detergents and micelles might stabilize certain structures of the protein core, but details of functional mechanism might be perturbed. There is no discussion of this assumption in the manuscript and the proteins environment is only briefly alluded to if you read the methods.

-Due to the large CSA effects of ^{19}F NMR caused linewidth broadening and the difficulty of expressing the GPCRs, it is still challenging to perform such experiments in the native membrane systems for membrane proteins like GPCRs. So far, two systems—micelle and HDL reconstitution systems are usually used for resolving the structures of GPCRs and other biophysical studies such as NMR, in which a vast majority of structures were resolved in micelles, mainly including two relatively stable detergent reconstitution systems DDM/CHS and MNG-3/CHS. In recent years, HDL systems were introduced into the GPCRs, and some structures have been resolved in different HDL systems. However, discrepancies between the HDL reconstituted GPCRs and pharmacological evaluations still exist, and the ratio of lipids significantly affects the functionality of the receptors, though we agree that HDL reconstitution may be a more biologically relevant system than detergent micelles in the best scenario for both systems. However, in both $\beta_2\text{AR}$ and $\text{A}_{2\text{A}}\text{R}$ systems subjected to conformational state study using ^{19}F NMR, the receptors were over-activated, reflected by an overwhelming portion of active states, as shown in the above Figure in response to Reviewer #1, while the subpopulation of the activated states for the MNG-3 reconstituted receptors was more aligned with pharmacological signaling, especially when one considers the $\beta_2\text{AR}$ is a low constitutive receptor. From this perspective, MNG-3/CHS may be a better system for studying the receptor activation process while the HDL system may be a better system for investigating the effects of lipids on the receptor activation and a better system for obtaining fully activated structures for cryo-EM as well because of the large portion of activated states presented in the HDL system. From this standpoint, we don't know yet which system, MNG-

3 or HDL systems, is better for resolving the intermediate complex structures because the HDL system tends to shift the conformational equilibrium to the fully activated states. We are examining these system-dependent behaviors in the lab currently.

In order to emphasize that the use of detergent micelles may affect functional mechanisms, we have added the following text to the Discussion section on pages 23:

It is also worthy of note that because ^{19}F profiles reveal that state populations and exchange dynamics depend on the environment (e.g., detergent micelle vs. lipid nanodiscs), additional studies are necessary to assess the relative importance of the intermediate states of the $A_{2A}\text{R}$, such as the S4 in lipidic environments and if it can be successfully populated in it as well, considering that the HDL systems prefer to shift the conformational equilibrium to the fully activated states as aforementioned^{4,19}.

With regard to methodological clarity, we believe that the text modifications noted in response to the previous criticism are sufficient to ensure that the reader is aware of the receptor's environment in both NMR and MD simulations and the reason we chose the MNG-3 system in this study, as described on page 7.

Thus, the receptors in this manuscript will be reconstituted in the MNG-3/CHS system^{4,28}. MNG-3/CHS is the most reliable reconstitution system for a GPCR structural biology study so far²⁹. To decrease the usage of receptor quantity for the NMR studies, MNG-3/CHS is also a better choice because an unavoidable disadvantage of using other reconstitution systems like high-density lipoprotein systems (HDL) is that they will cause a significant decrease in the final productivity of the receptors due to secondary reconstitution incorporation²⁸⁻³¹. The most recent studies in both $A_{2A}\text{R}$ and $\beta_2\text{AR}$ also indicated that HDL reconstituted receptors have an unreasonably high activity compared to the MNG-3/CHS reconstituted receptors. This unexpected high activity of HDL reconstituted receptors was not consistent with pharmacological studies, considering that $\beta_2\text{AR}$ is a low constitutive receptor and $A_{2A}\text{R}$ doesn't have such a high activity as well^{4,19}. The reason behind this phenomenon is still unclear, partially due to the limitation of the current HDL system, compared to a real physiological bilayer lipid environment. Therefore, MNG-3/CHS is a more desirable system for studying receptor conformational transitions and dynamics in activation process to avoid the effects from the HDL system while the HDL system could be better to study the effects of lipids on the receptor activations and structural elucidation for the fully activated complexes until a more desirable HDL system is available⁷.

2. Simulation Results: The simulation aspect of this manuscript is quite limited. There is single extended data figure 4 that shows how certain mutants behave differently from the WT. The description of these results should be providing the reader with a bit more detail beyond the results being consistent with ion lock breakage and helix 6 rotation. Please guide the reader the quantitative results that show this.

-We accordingly added more descriptions and discussions for the MD simulations with the revised Fig.1, part of Fig.4, Extended Data Figs. 4,5,10 and 11. The corresponding descriptions were also added to the manuscript. Furthermore, we have increased the duration of all MD simulations from 400 ns to 1 μs and updated all figures accordingly.

3. Simulation setup: As with the wetlab work, the authors decide to simulate a micelle. This is disappointing in that it would be nice to see if there is an effect to structure/function in the more native membrane environment. Moreover, the selection of micelle size is not clear. Why were 90 MNG-3 lipids decided to represent the micelle? Is the micelle that covers the GPCR in experiment this size or different? Is this micelle spherical or asymmetrical and by how much? How does the lipid headgroup interact with the cyto/extracellular parts of the GPCR? Could this alter structure and function?

-The selection of 90 MNG-3 lipids per micelle is now described in more detail in the Methods section on page 30, lines 458- 463:

Detergent micelles (as opposed to lipid bilayers) were selected for consistency with experimental data. Taking the molecular weights of A_{2A}R and MNG-3 to be 45 and 1 kDa, respectively, we reasoned that 90 MNG-3 detergents per micelle would approximate the 100-140 kDa apparent molecular weight of MNG-3/CHS reconstituted A_{2A}R measured previously by native PAGE ¹. This composition is similar to the 96 MNG-3 detergents per micelle used in a recent simulation study of the A_{2A}R and β₂AR ³².

To address questions about micelle shape and detergent protein interactions, we have added new Extended Data Figure 5: Detergent coverage of A_{2A}R in MD simulations (page 13). To address the question about whether detergents vs. lipids could alter structure and function, we refer the Reviewer to our response to their general comment #1, which we believe clarifies that this is possible without overdrawing conclusions in the absence of nanodisc simulations.

References:

- 1 Ye, L., Van Eps, N., Zimmer, M., Ernst, O. P. & Prosser, R. S. Activation of the A_{2A} adenosine G-protein-coupled receptor by conformational selection. *Nature* **533**, 265-268, doi:10.1038/nature17668 (2016).
- 2 Manglik, A. *et al.* Structural insights into the dynamic process of β_2 -adrenergic receptor signaling. *Cell* **161**, 1101-1111, doi:10.1016/j.cell.2015.04.043 (2015).
- 3 Ye, L. *et al.* Mechanistic insights into allosteric regulation of the A_{2A} adenosine G protein-coupled receptor by physiological cations. *Nat. Commun.* **9**, 1372, doi:10.1038/s41467-018-03314-9 (2018).
- 4 Huang, S. K. *et al.* Delineating the conformational landscape of the adenosine A_{2A} receptor during G protein coupling. *Cell* **184**, 1884-1894 e1814, doi:10.1016/j.cell.2021.02.041 (2021).
- 5 Prosser, R. S., Ye, L., Pandey, A. & Oraziotti, A. Activation processes in ligand-activated G protein-coupled receptors: A case study of the adenosine A_{2A} receptor. *Bioessays* **39**, doi:10.1002/bies.201700072 (2017).
- 6 Wang, X. *et al.* Trifluorinated keto-enol tautomeric switch in probing domain rotation of a G protein-coupled receptor. *Bioconjugate Chem.* **32**, 99-105, doi:10.1021/acs.bioconjchem.0c00670 (2021).
- 7 Ye, L., Wang, X., McFarland, A. & Madsen, J. J. ¹⁹F NMR: A promising tool for dynamic conformational studies of G protein-coupled receptors. *Structure*, doi:10.1016/j.str.2022.08.007 (2022).
- 8 Susac, L., Eddy, M. T., Didenko, T., Stevens, R. C. & Wuthrich, K. A_{2A} adenosine receptor functional states characterized by ¹⁹F-NMR. *Proc. Natl. Acad. Sci. U.S.A.* **115**, 12733-12738, doi:10.1073/pnas.1813649115 (2018).
- 9 Liu, J. J., Horst, R., Katritch, V., Stevens, R. C. & Wuthrich, K. Biased signaling pathways in β_2 -adrenergic receptor characterized by ¹⁹F-NMR. *Science* **335**, 1106-1110, doi:10.1126/science.1215802 (2012).
- 10 Susac, L., O'Connor, C., Stevens, R. C. & Wuthrich, K. In-Membrane Chemical Modification (IMCM) for Site-Specific Chromophore Labeling of GPCRs. *Angew Chem Int Ed Engl* **54**, 15246-15249, doi:10.1002/anie.201508506 (2015).
- 11 Kim, T. H. *et al.* The role of ligands on the equilibria between functional states of a G protein-coupled receptor. *J Am Chem Soc* **135**, 9465-9474, doi:10.1021/ja404305k (2013).
- 12 Yu, J.-X., Hallac, R. R., Chiguru, S. & Mason, R. P. New frontiers and developing applications in ¹⁹F NMR. *Prog. Nucl. Magn. Reson. Spectrosc.* **70**, 25-49, doi:<https://doi.org/10.1016/j.pnmrs.2012.10.001> (2013).
- 13 Wang, X. *et al.* A genetically encoded F-19 NMR probe reveals the allosteric modulation mechanism of cannabinoid receptor 1. *J. Am. Chem. Soc.* **143**, 16320-16325, doi:10.1021/jacs.1c06847 (2021).
- 14 Klein-Seetharaman, J., Getmanova, E. V., Loewen, M. C., Reeves, P. J. & Khorana, H. G. NMR spectroscopy in studies of light-induced structural changes in mammalian rhodopsin: applicability of solution ¹⁹F NMR. *Proc. Natl. Acad. Sci. U.S.A.* **96**, 13744-13749, doi:10.1073/pnas.96.24.13744 (1999).
- 15 Gerig, J. T. Fluorine NMR of proteins. *Prog. Nucl. Magn. Reson. Spectrosc.* **26**, 293-370 (1994).

- 16 Buchholz, C. R. & Pomerantz, W. C. K. ^{19}F NMR viewed through two different lenses: ligand-observed and protein-observed ^{19}F NMR applications for fragment-based drug discovery. *RSC Chem. Biol.* **2**, 1312-1330, doi:10.1039/d1cb00085c (2021).
- 17 Hull, W. E. & Sykes, B. D. Fluorine-19 nuclear magnetic resonance study of fluorotyrosine alkaline phosphatase: the influence of zinc on protein structure and a conformational change induced by phosphate binding. *Biochemistry* **15**, 1535-1546, doi:10.1021/bi00652a027 (1976).
- 18 Mafi, A., Kim, S. K. & Goddard, W. A. The mechanism for ligand activation of the GPCR-G protein complex. *Proc. Natl. Acad. Sci. U.S.A.* **119**, e2110085119, doi:10.1073/pnas.2110085119 (2022).
- 19 Staus, D. P., Wingler, L. M., Pichugin, D., Prosser, R. S. & Lefkowitz, R. J. Detergent- and phospholipid-based reconstitution systems have differential effects on constitutive activity of G-protein-coupled receptors. *J. Biol. Chem.* **294**, 13218-13223, doi:10.1074/jbc.AC119.009848 (2019).
- 20 Lerch, M. T. *et al.* Viewing rare conformations of the β_2 adrenergic receptor with pressure-resolved DEER spectroscopy. *Proc. Natl. Acad. Sci.* **117**, 31824-31831, doi:10.1073/pnas.2013904117 (2020).
- 21 Sykes, B. D., Weingarten, H. I. & Schlesinger, M. J. Fluorotyrosine alkaline phosphatase from *Escherichia coli*: preparation, properties, and fluorine-19 nuclear magnetic resonance spectrum. *Proc. Natl. Acad. Sci. U.S.A.* **71**, 469-473, doi:10.1073/pnas.71.2.469 (1974).
- 22 Mark A. Danielson & Falke, J. J. Use of ^{19}F NMR to probe protein structure and conformational changes. *Annu. Rev. Biophys.* **25**, 163-195 (1996).
- 23 Carpenter, B. & Tate, C. G. Active state structures of G protein-coupled receptors highlight the similarities and differences in the G protein and arrestin coupling interfaces. *Curr. Opin. Struct. Biol.* **45**, 124-132, doi:10.1016/j.sbi.2017.04.010 (2017).
- 24 Altenbach, C., Kusnetzow, A. K., Ernst, O. P., Hofmann, K. P. & Hubbell, W. L. High-resolution distance mapping in rhodopsin reveals the pattern of helix movement due to activation. *Proc. Natl. Acad. Sci. U.S.A.* **105**, 7439-7444, doi:10.1073/pnas.0802515105 (2008).
- 25 Dijkman, P. M. *et al.* Conformational dynamics of a G protein-coupled receptor helix 8 in lipid membranes. *Sci. Adv.* **6**, eaav8207, doi:10.1126/sciadv.aav8207 (2020).
- 26 Fay, J. F. & Farrens, D. L. Structural dynamics and energetics underlying allosteric inactivation of the cannabinoid receptor CB₁. *Proc. Natl. Acad. Sci. U.S.A.* **112**, 8469-8474, doi:10.1073/pnas.1500895112 (2015).
- 27 Nygaard, R. *et al.* The dynamic process of β_2 -adrenergic receptor activation. *Cell* **152**, 532-542, doi:10.1016/j.cell.2013.01.008 (2013).
- 28 Hagn, F., Etzkorn, M., Raschle, T. & Wagner, G. Optimized phospholipid bilayer nanodiscs facilitate high-resolution structure determination of membrane proteins. *J. Am. Chem. Soc.* **135**, 1919-1925, doi:10.1021/ja310901f (2013).
- 29 Chae, P. S. *et al.* Maltose-neopentyl glycol (MNG) amphiphiles for solubilization, stabilization and crystallization of membrane proteins. *Nat. Methods* **7**, 1003-1008, doi:10.1038/nmeth.1526 (2010).

- 30 Denisov, I. G., Grinkova, Y. V., Lazarides, A. A. & Sligar, S. G. Directed self-assembly of monodisperse phospholipid bilayer Nanodiscs with controlled size. *J. Am. Chem. Soc.* **126**, 3477-3487, doi:10.1021/ja0393574 (2004).
- 31 Nasr, M. L. *et al.* Covalently circularized nanodiscs for studying membrane proteins and viral entry. *Nat. Methods* **14**, 49-52, doi:10.1038/nmeth.4079 (2017).
- 32 Lee, S. *et al.* How do branched detergents stabilize GPCRs in micelles? *Biochemistry* **59**, 2125-2134, doi:10.1021/acs.biochem.0c00183 (2020).

Reviewers' Comments:

Reviewer #1:

Remarks to the Author:

In this revised version, there is a real effort of pedagogy but also a part of self-criticism concerning the method to investigate a space of conformations of a complex dynamic protein like a GPCR by a single fluorine probe out of the trans-membrane region with the help of *in silico* simulations. There are also additional comments regarding what has been done from other groups based on a similar approach, and this helps in particular to understand the differences observed between these studies. The authors have also added relevant data such as T2 relaxation experiments and GTPase hydrolysis experiments. So, overall, the manuscript has gained in clarity.

There is no doubt that this laboratory has a great expertise in biochemistry to prepare top quality fluorine-labeled receptors for NMR studies in detergent solutions and that the data displayed here in these conditions is relevant. However, for me there is still a major point that needs to be solved that concerns the argument developed to justify the fact of working in detergent solutions rather than in nanometric lipid bilayers.

The reason why a MNG detergent solution is more appropriate to study A2AR than lipid nanodiscs is not backed by sound arguments (including the answer addressed to the second reviewer). As mentioned in Staus et al (JBC, 394, 13218), « The contrast between the low levels of β 2AR constitutive activity in cells and the high constitutive activity observed in an isolated phospholipid bilayer indicates that β 2AR basal activity depends on the reconstitution system and further suggests that various cellular mechanisms suppress β 2AR basal activity physiologically. » I think that the argumentation displayed page 7 of the new version of the manuscript should be removed (lines 108 to 125) because this depreciates the quality of this manuscript.

A plethora of comparative *in vitro* studies performed with membrane proteins in detergent solutions versus lipid nanodiscs at a functional or structural levels logically indicate that a lipid bilayer environment is much more suitable. And this concerns all types of membrane proteins, procaryotic or eucaryotic, beta barrels or bundle of trans-membrane alpha-helices (e.g. see for instance Nature Chemical Biology | VOL 14 | JULY 2018 | 715–722), monomeric or oligomeric.

HDL or nanodiscs still do not perfectly mimic the native membrane where the receptor is supposed to be (phospholipid composition, ion concentrations, interacting proteins,...). Nobody is perfect and for sure there is still a gap between a nanodisc and a complex biological membrane. But it even becomes a question of common sense, i.e. a nanometric bilayer environment more closely mimics the physical properties of a biological membrane than a detergent micelle at the interface with the membrane protein. In addition, it appears, and A2AR represents a good example, that the conformational ensemble is very sensitive to the type and concentration of the detergent used, as perfectly mentioned by the authors in their answer.

However, probably 99.9% of *in vitro* studies of membrane proteins are performed in detergent solutions. So, it is still perfectly relevant to choose that type of surfactant for biophysical investigations. But, in the special case of A2AR + fluorine probe + solution state NMR, the study of Huang et al (Cell 184, 1884–1894) of A2AR in a lipid nanodiscs published in 2021 represents a major contribution -or a breakthrough- in the field and clearly and undeniably reshuffles the cards. I would have thought that the main argument to work with detergents associated with solution-state NMR spectroscopy is to work with faster tumbling objects compared to other membrane mimetics but it looks like that thanks to the slower exchange dynamics, Huang et al., noticed “an overall improved spectral resolution”.

To summarize all of this, the model presented here is new in the field, all the science seems to me perfectly done by a lab expert in the different technologies used here. I am also well aware of the

amount of work involved for such a very challenging subject. But, the question remains whether it represents a major new contribution to a wide readership like that of Nat Commun.

additional comments :

regarding my comment "only an aqueous residue was labeled": I can understand that such labeling is not feasible in the transmembrane part of the receptor without perturbing the functionality and/or stability of the protein. My comment was more about would it be profitable to associate to these experiments another labeling to complete the sampling of the conformation space and in particular to get rid of the dependence of a single probe on the physico-chemical conditions or the preparation of the sample?

Regarding my comment on partial deuteration of the receptor: Maybe my comment was not so clear. I was not talking about solvent isotope effect but just about reducing the dipolar interactions around the fluorine probes by switching ^1H by deuterons in the proteins thanks to the possibility to deuterate proteins using yeast as an expression host.

Reviewer #2:

Remarks to the Author:

the updated manuscript and its response to my past review is an improvement. The additional details of the methods and extending the MD simulations with more data makes this worthy of publishing.

POINT-TO-POINT RESPONSES TO REVIEWERS' COMMENTS (2nd Revision)

We thank the Reviewers for the further comments on the manuscript. Each related criticism is addressed below.

Response to Reviewer #1:

Reviewer #1 (Remarks to the Author):

In this revised version, there is a real effort of pedagogy but also a part of self-criticism concerning the method to investigate a space of conformations of a complex dynamic protein like a GPCR by a single fluorine probe out of the trans-membrane region with the help of in silico simulations. There are also additional comments regarding what has been done from other groups based on a similar approach, and this helps in particular to understand the differences observed between these studies. The authors have also added relevant data such as T2 relaxation experiments and GTPase hydrolysis experiments. So, overall, the manuscript has gained in clarity.

-We are grateful for recognizing our efforts for the revisions.

There is no doubt that this laboratory has a great expertise in biochemistry to prepare top quality fluorine-labeled receptors for NMR studies in detergent solutions and that the data displayed here in these conditions is relevant. However, for me there is still a major point that needs to be solved that concerns the argument developed to justify the fact of working in detergent solutions rather than in nanometric lipid bilayers.

The reason why a MNG detergent solution is more appropriate to study A2AR than lipid nanodiscs is not backed by sound arguments (including the answer addressed to the second reviewer). As mentioned in Staus et al (JBC, 394, 13218), « The contrast between the low levels of β 2AR constitutive activity in cells and the high constitutive activity observed in an isolated phospholipid bilayer indicates that β 2AR basal activity depends on the reconstitution system and further suggests that various cellular mechanisms suppress β 2AR basal activity physiologically. » I think that the argumentation displayed page 7 of the new version of the manuscript should be removed (lines 108 to 125) because this depreciates the quality of this manuscript.

-Thanks for bringing this up to our attention. As advised by the reviewer, we have removed related descriptions in this new version.

A plethora of comparative in vitro studies performed with membrane proteins in detergent solutions versus lipid nanodiscs at a functional or structural levels logically indicate that a lipid bilayer environment is much more suitable. And this concerns all types of membrane proteins, prokaryotic or eukaryotic, beta barrels or bundle of trans-membrane alpha-helices (e.g. see for instance Nature Chemical Biology | VOL 14 | JULY 2018 | 715–722), monomeric or oligomeric.

HDL or nanodiscs still do not perfectly mimic the native membrane where the receptor is supposed to be (phospholipid composition, ion concentrations, interacting proteins,...). Nobody is perfect and for sure there is still a gap between a nanodisc and a complex biological membrane. But it even becomes a question of common sense, i.e. a nanometric bilayer environment more closely mimics the physical properties of a biological membrane than a detergent micelle at the interface with the membrane protein. In addition, it appears, and A2AR represents a good example, that the conformational ensemble is very sensitive to the type and concentration of the detergent used, as perfectly mentioned by the authors in their answer.

-Thanks, we fully agree with the reviewer.

However, probably 99.9% of in vitro studies of membrane proteins are performed in detergent solutions. So, it is still perfectly relevant to choose that type of surfactant for biophysical investigations. But, in the special case of A2AR + fluorine probe + solution state NMR, the study of Huang et al (Cell 184, 1884–1894) of A2AR in a lipid nanodiscs published in 2021 represents a major contribution -or a breakthrough- in the field and clearly and undeniably reshuffles the cards. I would have thought that the main argument to work with detergents associated with solution-state NMR spectroscopy is to work with faster tumbling objects compared to other membrane mimetics but it looks like that thanks to the slower exchange dynamics, Huang et al., noticed “an overall improved spectral resolution”.

-Thank the reviewer for the deep insights in both GPCR and NMR. We appreciate these comments, which will help the field development by further improving receptor conformational resolution through various innovations, including the reconstitution systems that can best represent the native membrane.

To summarize all of this, the model presented here is new in the field, all the science seems to me perfectly done by a lab expert in the different technologies used here. I am also well aware of the amount of work involved for such a very challenging subject. But, the question remains whether it represents a major new contribution to a wide readership like that of Nat Commun.

-We thank for constructive feedback from the reviewer, particularly those towards NMR data processing and the model of GPCR activation proposed in the previous version. The feedback has substantially improved the quality of our manuscript.

additional comments :

regarding my comment “only an aqueous residue was labeled”: I can understand that such labeling is not feasible in the transmembrane part of the receptor without perturbing the functionality and/or stability of the protein. My comment was more about would it be profitable to associate to these experiments another labeling to complete the sampling of

the conformation space and in particular to get rid of the dependence of a single probe on the physico-chemical conditions or the preparation of the sample?

-It is an excellent suggestion to expand one site labeling to multisites for a complete sampling of any possible conformation space of a receptor. To be honest, any site selection that is able to represent a set of conformational states in a sufficient resolution is not easy for any given membrane protein system like GPCRs. As mentioned in our previous revision, our lab is making all efforts to label non-solvent exposure regions (including transmembrane domains) using unnatural amino acid genetic incorporation with the possibility of profiling conformational space of transmembrane domains. The success of these strategies by labeling multi-sites should provide a conformational matrix to get rid of the dependence of a single probe. We wish to make some substantial progress in the near future.

Regarding my comment on partial deuteration of the receptor: Maybe my comment was not so clear. I was not talking about solvent isotope effect but just about reducing the dipolar interactions around the fluorine probes by switching ^1H by deuterons in the proteins thanks to the possibility to deuterate proteins using yeast as an expression host.

-We appreciate that the reviewer proposed the idea to replace the ^1H with ^2H using deuterated culture, which could reduce the space-dependent dipolar interactions around the fluorine probe. We are sorry that we couldn't offer a positive response at this stage because deuteration of media and chemicals in a large expression system is not an easy task for a small lab like ours, considering the costs of the budget and personnel. As it is well-known that the difficulty of GPCR expression itself and the deuteration will exaggerate these expressions, though the related yeast expressions have been performed in several labs in recent years for ^{13}C - or ^{15}N -based deuterated NMR experiments, especially for TROSY experiments¹⁻⁴. There is no report so far in regard to the deuterated system in reducing ^1H - ^{19}F space interactions, which should in principle benefit to the decrease of linewidth broadening. On the other hand, a limitation of deuteration by expression of GPCRs in *P. pastoris* grown in D_2O media has been reported, in which the back-protonation of amide groups may be incomplete⁵. This deficit could lead to the complexity of space dependent ^1H - ^{19}F dipolar interactions. We appreciate the valuable suggestion and look forward to integrating it into our future research and further advance the resolution of conformational states.

Response to Reviewer #2:

Reviewer #2 (Remarks to the Author):

the updated manuscript and its response to my past review is an improvement. The additional details of the methods and extending the MD simulations with more data makes this worthy of publishing.

-Thanks so much for the positive responses to our revisions.

References

- 1 Clark, L. D. *et al.* Ligand modulation of sidechain dynamics in a wild-type human GPCR. *Elife* **6**, doi:10.7554/eLife.28505 (2017).
- 2 Eddy, M. T. *et al.* Allosteric coupling of drug binding and intracellular signaling in the A_{2A} adenosine receptor. *Cell* **172**, 68-80 e12, doi:10.1016/j.cell.2017.12.004 (2018).
- 3 Clark, L. *et al.* Methyl labeling and TROSY NMR spectroscopy of proteins expressed in the eukaryote *Pichia pastoris*. *J Biomol NMR* **62**, 239-245, doi:10.1007/s10858-015-9939-2 (2015).
- 4 Eddy, M. T., Martin, B. T. & Wuthrich, K. A_{2A} adenosine receptor partial agonism related to structural rearrangements in an activation microswitch. *Structure* **29**, 170-176 e173, doi:10.1016/j.str.2020.11.005 (2021).
- 5 Shimada, I., Ueda, T., Kofuku, Y., Eddy, M. T. & Wuthrich, K. GPCR drug discovery: integrating solution NMR data with crystal and cryo-EM structures. *Nat Rev Drug Discov* **18**, 59-82, doi:10.1038/nrd.2018.180 (2019).